# Towards a More Rigorous Science of Blindspot Discovery in Image Classification Models

**Gregory Plumb**[†*1, 2], **Nari Johnson**[†1], **Ángel Alexander Cabrera**[1], **& Ameet Talwalkar**[1]
[1]**Carnegie Mellon University**
[2]**Inpleo, Inc.**
`{gdplumb,narij}@andrew.cmu.edu`

Reviewed on OpenReview: `https://openreview.net/forum?id=MaDvbLaBiF`

## Abstract

A growing body of work studies Blindspot Discovery Methods (BDMs): methods that use an image embedding to find semantically meaningful (*i.e.,* united by a human-understandable concept) subsets of the data where an image classifier performs significantly worse. Motivated by gaps in prior work, we introduce a new framework for evaluating BDMs, `SpotCheck`, that uses synthetic image datasets to train models with known blindspots and a new BDM, `PlaneSpot`, that uses a 2D image representation. We use `SpotCheck` to run controlled experiments that identify factors that influence BDM performance (*e.g.,* the number of blindspots in a model, or features used to define the blindspot) and show that `PlaneSpot` is competitive with and in many cases outperforms existing BDMs. Importantly, we validate these findings by designing additional experiments that use real image data from MS-COCO, a large image benchmark dataset. Our findings suggest several promising directions for future work on BDM design and evaluation. Overall, we hope that the methodology and analyses presented in this work will help facilitate a more rigorous science of blindspot discovery.

## 1 Introduction

A growing body of work has found that models with high average test performance can still make systemic errors, which occur when the model performs significantly worse on a semantically meaningful (*i.e.,* united by a human-understandable concept) subset of the data (Buolamwini & Gebru, 2018; Chung et al., 2019; Oakden-Rayner et al., 2020; Singla et al., 2021; Ribeiro & Lundberg, 2022). For example, past works have demonstrated that models trained to diagnose skin cancer from dermoscopic images sometimes rely on spurious artifacts (*e.g.,* surgical skin markers that some dermatologists use to mark lesions); consequently, they have different performance on images with or without those spurious artifacts (Winkler et al., 2019; Mahmood et al., 2021). More broadly, finding systemic errors can help us detect algorithmic bias (Buolamwini & Gebru, 2018) or sensitivity to distribution shifts (Sagawa et al., 2020; Singh et al., 2020).

In this work, we focus on what we call the *blindspot discovery* problem, which is the problem of finding an image classification model's systemic errors without making many of the assumptions considered in related works (*e.g.,* we do not assume access to metadata to define semantically meaningful subsets of the data, tools to produce counterfactual images, a specific model structure or training process, or a human in the loop). We call methods for addressing this problem Blindspot Discovery Methods (BDMs) (*e.g.,* Kim et al., 2019; Sohoni et al., 2020; Singla et al., 2021; d'Eon et al., 2021; Eyuboglu et al., 2022).

We note that blindspot discovery is an emerging research area and that there has been more emphasis on developing new BDMs than on formalizing the problem itself. Consequently, we propose a problem formalization, summarize different approaches for evaluating BDMs, and summarize several high-level design

---

*The author completed the majority of the work as a student at CMU.
†Equal Contribution

choices made by BDMs. When we do this, we observe the following two gaps. First, existing evaluations are based on an incomplete knowledge of the model's blindspots, which limits the types of measurements and claims they can make. Second, dimensionality reduction is a relatively underexplored aspect of BDM design.

Motivated these gaps in prior work, we propose a new evaluation framework, `SpotCheck`, and a new BDM, `PlaneSpot`. `SpotCheck` is a synthetic evaluation framework for BDMs that differs from past evaluations in that it gives us complete knowledge of the model's blindspots and allows us to identify factors that influence BDM performance. Additionally, we refine the evaluation metrics used by prior work. Inspired by the intuition that clustering is typically easier in lower dimensions, we introduce `PlaneSpot`, a simple BDM that finds blindspots by clustering on a low-dimensional 2D image representation.

We use `SpotCheck` to run controlled experiments to identify several factors that influence BDM performance. We run additional experiments using $100,000$ photographs from the MS-COCO dataset (Lin et al., 2014), a large-scale object detection benchmark, and find that these trends discovered using `SpotCheck` generalize. Our experiments show that `PlaneSpot` is competitive with and in many cases outperforms existing BDMs, a finding that has exciting implications for future work on interactive blindspot discovery. In our extended discussion, we present several promising directions for future research on BDMs to address the failure modes that we discovered. Overall, we hope that the methodology and analyses presented in this work will help facilitate a more rigorous science of blindspot discovery.

## 2 Background

In this section, we formalize the problem of *blindspot discovery* for image classification. We then discuss general approaches for evaluating the Blindspot Discovery Methods (BDMs) designed to address this problem and high-level design choices made by BDMs.

**Problem Definition.** The broad goal of finding systemic errors has been studied across a range of problem statements and method assumptions. Some common assumptions are:

- Access to metadata help define coherent subsets of the data (*e.g.,* Kim et al., 2018; Buolamwini & Gebru, 2018; Singh et al., 2020).
- The ability to produce counterfactual images (*e.g.,* Shetty et al., 2019; Singla et al., 2020; Xiao et al., 2021; Leclerc et al., 2021; Li & Xu, 2021; Lang et al., 2021; Plumb et al., 2022; Wiles et al., 2023).
- A human-in-the loop, either through an interactive interface (*e.g.,* Cabrera et al., 2019; Balayn et al., 2022; Rajani et al., 2022; Gao et al., 2022; Cabrera et al., 2023; Suresh et al., 2023) or by inspecting explanations (*e.g.,* Yeh et al., 2020; Adebayo et al., 2022).

While appropriate at times, these assumptions all restrict the applicability of their respective methods. For example, consider assuming access to metadata to help define coherent subsets of the data. This metadata is much less common in applied settings than it is for common ML benchmarks. Further, the efficacy of methods that rely on this metadata is limited by the quantity and relevance of this metadata; in general, efficiently collecting large quantities of relevant metadata is challenging because it requires that the model developer can anticipate all possible relevant types of systemic errors.

Consequently, we define the problem of blindspot discovery as the problem of finding an image classification model's systemic errors without making any of these assumptions. More formally, suppose that we have an image classifier, $f$, and a dataset of labeled images, $D = [x_i]_{i=1}^n$. Then, a *blindspot* is a coherent (*i.e.,* semantically meaningful, or united by a human-understandable concept) set of images, $\Psi \subset D$, where $f$ performs significantly worse (*i.e.,* $p(f, \Psi) \ll p(f, D \setminus \Psi)$ for some performance metric, $p$, such as recall). We denote the set of $f$'s *true blindspots* as $\mathbf{\Psi} : \{\Psi_m\}_{m=1}^M$. Next, we define the problem of *blindspot discovery* as the problem of finding $\mathbf{\Psi}$ using only $f$ and $D$. Then, a BDM is a method that takes as input $f$ and $D$ and outputs an ordered (by some definition of importance) list of *hypothesized blindspots*, $\hat{\mathbf{\Psi}} : [\hat{\Psi}_k]_{k=1}^K$. Note that the $\Psi_m$ and $\hat{\Psi}_k$ are sets of images. Past works propose that human stakeholders can then inspect the output groups of points $\hat{\Psi}_k$ to describe the semantic features shared by each group, and that the $\hat{\Psi}_k$ can be given as input to algorithms that aim to "fix" the blindspot by learning an updated model (Sagawa et al., 2020).

**Approaches to BDM evaluation.** We observe that existing approaches to quantitatively evaluate BDMs fall in two categories. The first category of evaluations (Singla et al., 2021; d'Eon et al., 2021) simply measure the error rate or size of $\hat{\Psi}_k$. However, these evaluations have two problems. First, none of the properties they

| Method | 1. Image Representation | 2. Dimensionality Reduction | 3. Hypothesis Class |
|---|---|---|---|
| Multiaccuracy (Kim et al., 2019) | VAE representation | | Linear model |
| GEORGE (Sohoni et al., 2020) | Model representation | UMAP ($d = 0, 1, 2$) | Gaussian kernels |
| Spotlight (d'Eon et al., 2021) | Model representation | | Gaussian kernels |
| Barlow (Singla et al., 2021) | Adversarially-Robust Model representation | | Decision Tree |
| Domino (Eyuboglu et al., 2022) | CLIP representation | PCA ($d = 128$) | Gaussian kernels |
| `PlaneSpot` | Model representation | scvis ($d = 2$) | Gaussian kernels |

Table 1: A high level overview of the high-level design choices made by different BDMs.

measure capture whether $\hat{\Psi}_k$ is coherent (*e.g.,* a random sample of misclassified images has high error but may not match a single semantically meaningful description). Second, $f$'s performance on $\hat{\Psi}_k$ may not be representative of $f$'s performance on similar images because BDMs are optimized to return high error images (*e.g.,* suppose that $f$ has a 90% accuracy on images of "zebras with people"; then, by returning the 10% of such images that are misclassified, a BDM could mislead us into believing that $f$ has a 0% accuracy on *all* images of "zebras with people").

The second category of evaluations compares $\hat{\mathbf{\Psi}}$ to a subset of $\mathbf{\Psi}$ that has either been previously found or artificially induced (Sohoni et al., 2020; Eyuboglu et al., 2022). While these evaluations address several issues with those from the first category, they require knowledge of $\mathbf{\Psi}$, which is usually incomplete (*i.e.,* we usually do not know some of the $\Psi_m$). This incompleteness makes it difficult to identify factors that influence BDM performance or to measure a BDM's recall or false positive rate. It is not practical to fix this incompleteness using real data because we cannot realistically enumerate all of the possible coherent subsets of $D$ to check if they are blindspots. To address these limitations, we introduce `SpotCheck`, which gives us complete knowledge of $\mathbf{\Psi}$ by using synthetic data.

**High-level design choices of BDMs.** In Table 1, we summarize three of the high-level design choices made by existing BDMs. First, each BDM uses a model to extract an *image representation*. Many BDMs use a representation from $f$, but some use pre-trained external models or other models trained on $D$. Second, some of the BDMs apply some form of *dimensionality reduction* to that image representation. Third, each BDM learns a model from a specified *hypothesis class* to predict if an image belongs to a blindspot from that image's (potentially reduced) representation.

Interestingly, while there has been significant effort focused on the choice of a BDM's image representation and hypothesis class (along with its associated learning algorithm), we note that few past works discover blindspots using a low-dimensional representation.[1] This is surprising because these BDMs are all solving clustering or learning problems, which are generally easier in lower dimensions. Motivated by this gap in prior work and the visualization potential of a 2D representation, we introduce `PlaneSpot`.

## 3 Evaluating BDMs using Knowledge of the True Blindspots

In Section 3.1, we introduce `SpotCheck`, which is an evaluation framework for BDMs that gives us complete knowledge of the model's true blindspots by using synthetic images, and allows us to identify factors that influence BDM performance. In Section 3.2, we define the metrics that we use to measure the performance of BDMs given complete knowledge of the model's true blindspots.

### 3.1 `SpotCheck`: A Synthetic Evaluation Framework for BDMs

`SpotCheck` builds on ideas from Kim et al. (2022) by generating synthetic datasets of varying complexity and training models to have specific blindspots on those datasets. We summarize its key steps below; see Appendix A for details.

**Dataset Definition.** Each dataset generated by `SpotCheck` is defined using *semantic features* that describe the possible types of images it contains. All images generated by `SpotCheck` have a background and potentially multiple objects that have different shapes (*e.g.,* squares, rectangles, circles, or text). Each object in the

---

[1]GEORGE (Sohoni et al., 2020) does experiment with clustering on a low-dimensional representation. However, their approach differs from `PlaneSpot` in that they select a *different number* of UMAP components for different datasets. In contrast, we specifically study the effectiveness of clustering on a **2D** representation.

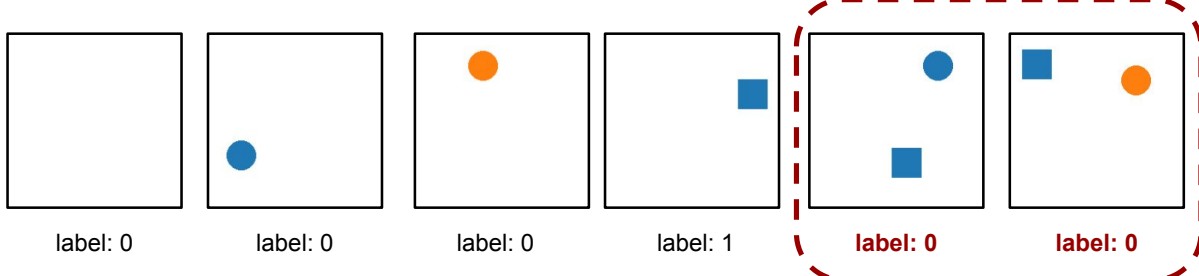

**blindspot: images with squares and circles**

Figure 1: By controlling the data generation process, `SpotCheck` gives us complete knowledge of a model's true blindspots. Here, we show images sampled from a dataset created with `SpotCheck`. **Dataset Complexity.** This dataset is defined by 3 semantic features that vary across images: the presence of a square, the presence of a circle, and the color of the circle. We do not count the "color of the square" because it is always blue. **Blindspot Specificity.** This blindspot is defined by 2 semantic features: the presence of a square and the presence of a circle. As a result, it contains any image with both a square and a circle, regardless of the circle's color. **Data Labels.** In general, the label indicates if a square is present. However, training images belonging to a blindspot are mislabeled.

image also has an associated size (large or small), color (blue or orange), and texture (solid or vertical stripes). Datasets that have a larger number of features have a larger variety of images and are therefore more complex. For example, a simple dataset may only contain images with squares and blue or orange circles (see Figure 1), while a more complex dataset may also contain images with striped rectangles, small text, or grey backgrounds. We detail the complete set of semantic features that `SpotCheck` uses in Appendix Section A.1.

**Blindspot Definition.** Each blindspot is defined using a subset of the semantic features that define its associated dataset (see Figure 1 for an example). Similarly to how a dataset with more features is more complex, a blindspot defined using more semantic features is more specific.

**Training a Model to have Specific Blindspots.** For each dataset and blindspot specification, we train a ResNet-18 model (He et al., 2016) to predict whether a square is present. To induce blindspots, we generate data where the label for each image in the training and validation sets is correct if and only if it does not belong to any of the blindspots (see Figure 1). The test set images are always correctly labeled. We chose to train using mislabeled images because it was most reliable method at inducing the desired true blindspots (see our extended discussion in Appendix B). Then, because we know the full set of semantic features that define the dataset, we can verify that the model learned to have exactly the set of blindspots that we intended it to. As a result, `SpotCheck` gives us *complete knowledge* of the model's true blindspots.

**Generating Diverse Experimental Configurations (ECs).** Since our goal is to study how various factors influence BDM performance, we generate a diverse set of *experimental configurations*, (*i.e.,* dataset, blindspots, and model triplets). To do this, we randomize several *factors*: the features that define a dataset (both the number of them and what they are) as well as the blindspots (the number of them, the number of features that define them, and what those features are). Importantly, we sample these factors independently of one another across this set of ECs so that we can estimate their individual influence on BDM performance.

## 3.2 Evaluation Metrics based on Knowledge of the True Blindspots

We define several evaluation metrics to measure how well the hypothesized blindspots returned by a BDM, $\hat{\mathbf{\Psi}}$, capture a model's true blindspots, $\mathbf{\Psi}$. Recall that each $\hat{\Psi}_k$ and $\Psi_m$ are sets of images. First, we measure how well a BDM finds each *individual* true blindspot (Blindspot Recall) and build on that to measure how well a BDM finds the *complete set* of true blindspots (Discovery Rate and False Discovery Rate). Our proposed

evaluation metrics refine those used in prior work, which only report the precision or recall of individual hypothesized blindspots (see detailed discussion in Appendix C).

**Blindspot Precision.** If $\hat{\Psi}_k$ is a subset of $\Psi_m$, we know that the model underperforms on $\hat{\Psi}_k$ and that $\hat{\Psi}_k$ is coherent. We measure this using the precision of $\hat{\Psi}_k$ with respect to $\Psi_m$:

$$\mathrm{BP}(\hat{\Psi}_k, \Psi_m) = \frac{|\hat{\Psi}_k \cap \Psi_m|}{|\hat{\Psi}_k|} \tag{1}$$

Then, we say that $\hat{\Psi}_k$ *belongs to* $\Psi_m$ if, for some threshold $\lambda_p$:

$$\mathrm{BP}(\hat{\Psi}_k, \Psi_m) \geq \lambda_p \tag{2}$$

However, $\hat{\Psi}_k$ can belong to $\Psi_m$ without capturing the same information as $\Psi_m$. For example, $\hat{\Psi}_k$ could be "zebras with people" while $\Psi_m$ could be "zebras (with or without people)". Because this excessive specificity could result in conclusions that are too narrow, we need to incorporate some notion of recall into the evaluation.

**Blindspot Recall.** One way to incorporate recall is using the proportion of $\Psi_m$ that $\hat{\Psi}_k$ covers:

$$\mathrm{BR}_{\mathrm{naive}}(\hat{\Psi}_k, \Psi_m) = \frac{|\hat{\Psi}_k \cap \Psi_m|}{|\Psi_m|} \tag{3}$$

We relax this definition by allowing $\Psi_m$ to be covered by the union of multiple $\hat{\Psi}_k$ that belong to it:

$$\mathrm{BR}(\hat{\boldsymbol{\Psi}}, \Psi_m) = \frac{\left| \left( \bigcup_{\hat{\Psi}_k : \mathrm{BP}(\hat{\Psi}_k, \Psi_m) \geq \lambda_p} \hat{\Psi}_k \right) \cap \Psi_m \right|}{|\Psi_m|} \tag{4}$$

Then, we say that $\hat{\boldsymbol{\Psi}}$ *covers* $\Psi_m$ if, for some threshold $\lambda_r$:

$$\mathrm{BR}(\hat{\boldsymbol{\Psi}}, \Psi_m) \geq \lambda_r \tag{5}$$

We do this because "zebras with people" and "zebras without people" belong to and jointly cover "zebras." So, if a BDM returns both, a user could combine them to arrive at the correct conclusion.

**Discovery Rate (DR).** We define the DR of $\hat{\boldsymbol{\Psi}}$ and $\boldsymbol{\Psi}$ as the fraction of the $\Psi_m$ that $\hat{\boldsymbol{\Psi}}$ covers:

$$\mathrm{DR}(\hat{\boldsymbol{\Psi}}, \boldsymbol{\Psi}) = \frac{1}{M} \sum_m \mathbb{1}(\mathrm{BR}(\hat{\boldsymbol{\Psi}}, \Psi_m) \geq \lambda_r) \tag{6}$$

**False Discovery Rate (FDR).** When the DR is non-zero, we define FDR of $\hat{\boldsymbol{\Psi}}$ and $\boldsymbol{\Psi}$ as the fraction of the $\hat{\Psi}_k$ that do not belong to any of the $\boldsymbol{\Psi}$:[2]

$$\mathrm{FDR}(\hat{\boldsymbol{\Psi}}, \boldsymbol{\Psi}) = \frac{1}{K} \sum_k \mathbb{1}(\max_m BP(\hat{\Psi}_k, \Psi_m) < \lambda_p) \tag{7}$$

Note that it is impossible to calculate the FDR without the complete set of true blindspots. While `SpotCheck` gives us this knowledge, it is generally not available for a model trained on an arbitrary image dataset.

---

[2]While calculating DR, we may only need the top-$u$ items of $\hat{\boldsymbol{\Psi}}$. As a result, we only calculate the FDR over those top-$u$ items. This prevents the FDR from being overly pessimistic when we intentionally pick $K$ too large in our experiments. However, it is unclear what value of $u$ to use for ECs that have a DR of zero, so we exclude these ECs from our FDR analysis.

## 4 `PlaneSpot`: **A simple BDM based on Dimensionality Reduction**

In this section, we define `PlaneSpot`, a new BDM. As shown in Table 1, `PlaneSpot` uses the most common choices for the image representation (*i.e., f*'s own representation) and the hypothesis class (*i.e.,* Gaussian kernels). `PlaneSpot` also uses standard techniques to learn a model from that hypothesis class. As a result, the most interesting aspect of `PlaneSpot`'s design is that it finds blindspots using a low-dimensional (2D) image representation. We start by defining some additional notation and then explain `PlaneSpot`'s choice for each of the high-level BDM design choices shown in Table 1.

**Notation.** Suppose that we want to find $f$'s blindspots for a class, $c$, and let $D^c$ be the set of images from $D$ that belong to $c$. Further, suppose that we have divided $f$ into two parts: $g : X \to \mathbb{R}^d$, which extracts $f$'s representation of an image (*i.e.,* its penultimate layer activations), and $h : \mathbb{R}^d \to [0, 1]$, which gives $f$'s predicted confidence for class $c$.

**Image Representation.** We use $g$ to extract $f$'s representation for $D^c$, $G = g(D^c) \in \mathbb{R}^{n \times d}$, and $h$ to extract $f$'s predicted confidences for class $c$, $H = h(G) \in [0, 1]^{n \times 1}$. Note that, because all of the images in $D^c$ belong to class $c$, entries of $H$ closer to 1 denote higher confidence in the class.

**Dimensionality Reduction.** We use $G$ to train scvis (Ding et al., 2018), which combines the objective functions of tSNE and an autoencoder, in order to learn a 2D representation of $f$'s representation, $s : \mathbb{R}^d \to \mathbb{R}^2$. We chose to use scvis because we believe there are implicit similarities between scvis's initial use (identifying cell types) and blindspot discovery: both tasks require finding potentially small groups of points in a high dimensional space (Ding et al., 2018). We also ablate our choice of dimensionality reduction method and find that scvis achieves comparable or better performance than other methods in Appendix D. We use $s$ to get the 2D representation of $D^c$, $S = s(G) \in \mathbb{R}^{n \times 2}$. Finally, we normalize the columns of $S$ to be in $[0, 1]$, $\bar{S} \in [0, 1]^{n \times 2}$.

**Hypothesis Class.** We want `PlaneSpot` to be aware of both $\bar{S}$ (the representation of $D^c$) and $H$ ($f$'s predicted confidences for $D^c$), so we combine them into a single representation, $R = [\bar{S}; w \cdot H] \in \mathbb{R}^{n \times 3}$ where $w$ is a hyperparameter controlling the relative weight of these components. We pass $R$ to a Gaussian Mixture Model clustering algorithm, where the number of clusters is chosen using the Bayesian Information Criteria. We return those clusters (where each $\hat{\Psi}_k$ corresponds to a different cluster) in order of decreasing importance, which we define using the product of their error rate and the number of errors in them.

## 5 Experiments

In Section 5.1, we use `SpotCheck` to run a series of controlled experiments using synthetic data. In Section 5.2, we demonstrate how to setup semi-controlled experiments to validate that the findings from `SpotCheck` generalize to settings with realistic image data.

### 5.1 Synthetic Data Experiments

We use `SpotCheck` to generate 100 ECs with datasets that have 6-8 semantic features and models that have 1-3 blindspots (each defined with 5-7 semantic features). We validate that the trained ResNet-18 models have near-perfect (99%) validation set accuracy on images that do not belong to a blindspot, and near-zero validation set accuracy on images that do belong to a blindspot to rule out the possibility of other coherent and underperforming subgroups beyond the blindspots we induced.

We evaluate three recent BDMs: Spotlight (d'Eon et al., 2021), Barlow (Singla et al., 2021), Domino (Eyuboglu et al., 2022), and `PlaneSpot`. We ran the BDMs on a held-out test set (instead of the model's train set) because the model's performance on the test set is likely to be more representative of model performance during deployment (Feldman & Zhang, 2020). We give all BDMs the positive examples (*i.e.,* images with squares) from the test set and limit them to returning 10 hypothesized blindspots. We use a held-out set of 20 ECs to tune each BDM's hyperparameters (see details in Appendix E). We use $\lambda_p = \lambda_r = 0.8$ for our metrics. We define each 95% confidence interval as the empirical mean plus or minus 1.96 standard deviations.

**Overall Results.** Table 2 shows the DR and FDR results averaged across all 100 ECs. We observe that `PlaneSpot` has the highest DR and that `PlaneSpot` and Barlow have a lower FDR than Spotlight and Domino. In Appendix F, we take a deeper look at why these BDMs are failing and conclude that a significant

Table 2: (*Synthetic Data*) Average BDM DR and FDR with their standard errors across 100 `SpotCheck` ECs.

| Method | DR | FDR |
|---|---|---|
| Barlow | 0.43 (0.04) | 0.03 (0.01) |
| Spotlight | 0.79 (0.03) | 0.09 (0.01) |
| Domino | 0.64 (0.04) | 0.07 (0.01) |
| **PlaneSpot** | **0.88 (0.03)** | **0.02 (0.01)** |

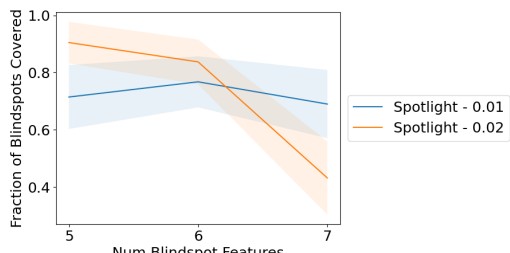

Figure 2: (*Synthetic Data*) The fraction of blindspots covered (with shaded 95% CIs) for blindspots defined using 5, 6, and 7 features using 2 different choices for the "minimum weight" hyperparameter $S$ (defined in d'Eon et al. (2021)).

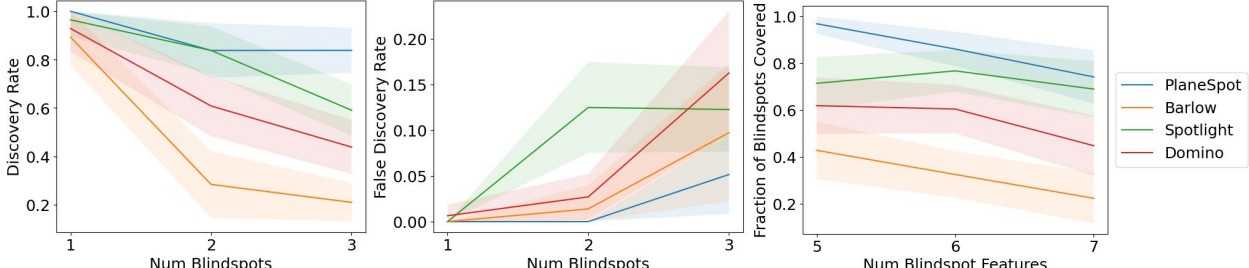

Figure 3: (*Synthetic Data*) (Left/Center) Average BDM DR/FDR (with shaded 95% confidence intervals) for ECs that have 1, 2, and 3 blindspots. (Right) The fraction of blindspots covered, across the individual blindspots from the ECs, for blindspots defined using 5, 6, and 7 features.

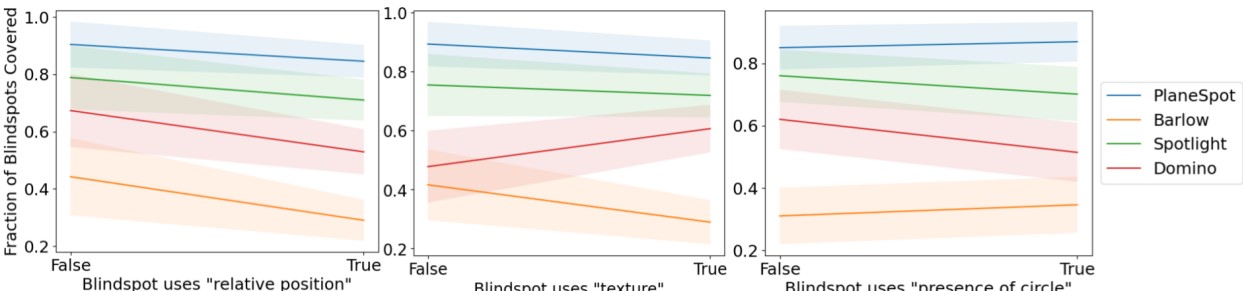

Figure 4: (*Synthetic Data*) The fraction of blindspots covered (with shaded 95% confidence intervals) that are or are not defined with (Left) the "relative position" feature (whether the square appears above the centerline of the image), (Center) a "texture" feature (whether any object in the image has vertical stripes), or (Right) the presence of a circle (whether there is a circle in the image).

portion of all of their failures can be explained by their tendency to merge multiple true blindspots into a single hypothesized blindspot.

### 5.1.1 Identifying factors that influence BDM performance

We study two types of factors: *per-configuration factors*, which measure properties of the dataset (*e.g.,* how complex is it?) or of the model (*e.g.,* how many blindspots does it have?), and *per-blindspot factors*, which measure properties of each individual blindspot (*e.g.,* is it defined with this feature?). For per-configuration factors, we average DR and FDR across the ECs. For per-blindspot factors, we report the fraction of blindspots covered averaged across each individual blindspot from the ECs (see Equation 5).

**The number of blindspots matters.** In Figure 3 (Left), we plot the average DR for ECs with $1, 2$, and $3$ blindspots. Average DR decreases for all methods as the number of blindspots increases. Figure 3 (Center) shows that FDR increases as the number of blindspots increases. Together these observations show that BDMs perform worse in settings with multiple blindspots, which is particularly significant because past evaluations have primarily focused on settings with one blindspot.

**The specificity of blindspots matters.** In Figure 3 (Right), we plot the fraction of blindspots covered for blindspots defined using $5, 6$, and $7$ features. With the exception of Spotlight, all of these methods are less capable of finding more-specific/less frequently occurring blindspots.

**The features that define a blindspot matter.** In Figure 4, we plot the fraction of blindspots covered for blindspots that either are or are not defined using various features. In general, we observe that the performance of these BDMs is influenced by the types of features used to define a blindspot (*e.g.,* the presence of spurious objects, color or texture information, background information). All methods are less likely to find blindspots defined using the "relative position" feature. Interestingly, BDMs that use the model's representation for their image representation (*i.e.,* `PlaneSpot` and Spotlight) are *less sensitive* to the features that define a blindspot than those that use an external model's representation (*i.e.,* Barlow and Domino).

**Hyperparameters matter.** We make two observations about BDM hyperparameters that suggest that it is critical that future BDMs provide ways to tune their hyperparameters. First, in Appendix E, we observe that many BDMs have significant performance differences when we vary their hyperparameters. However, hyperparameter tuning is harder in real settings where a model developer does not have access to information about the true blindspots. Second, in Figure 2, we observe that two hyperparameter settings that perform nearly identically on average, exhibit significantly different performance at finding blindspots defined using differing numbers of features. This suggests that there may not be a single best hyperparameter choice to find all of the blindspots in a single model, which could contain multiple blindspots of different specificity or size.

### 5.2 Real Data Experiments

We design semi-controlled experiments using the COCO dataset (Lin et al., 2014) to test whether several findings observed using `SpotCheck` generalize to settings with real data. While COCO has extensive annotations, it does not have enough metadata to test all of the findings from `SpotCheck` (*e.g.,* we cannot induce blindspots that depend on the texture of an object). Therefore, we study the following questions:

**Q1.** Will the 2D image representation used by `PlaneSpot` still be effective?
**Q2.** Are BDMs still less effective for models with more true blindspots?
**Q3.** Are BDMs still less effective at finding more-specific blindspots?

We are interested in studying the effect of two experimental factors: the *number* of blindspots in a model (Q2) and of the *specificity* of those blindspots (Q3). To estimate their influence on BDM performance, we use the same strategy as `SpotCheck` and generate a set of ECs where we randomize these factors independently of one another. We detail how we generate these ECs for reproducibility in Appendix G and show an example EC in Figure 5. We define each "more-specific" blindspot using two COCO object classes (*e.g.,* "elephants with people"), and each "less-specific" blindspot using only one class (*e.g.,* "zebras").

**blindspot 1: elephants with people**

label: 0      label: 1      label: 0      **label: 0**      **label: 0**      **label: 0**

**blindspot 2: zebras**

Figure 5: Images sampled from an example EC created using COCO. In this example, the task is to detect whether an animal is present. The EC has two true blindspots: "elephants with people," which is more-specific because it is defined using two objects, and "zebras," which is less-specific because it is defined using only one object.

Table 3: (*Real Data*) The effect of number of blindspots (Q2) is confounded with the effect of the specificity of those blindspots (Q3): as we try to induce more blindspots in the model, the model is less likely to learn more-specific blindspots. Overall, the model learned all of the intended blindspots in 65 of the 90 ECs in the general pool.

| Num Blindspots (Q2) | Fraction of successfully induced Blindspots that are more-specific (Q3) |
|---|---|
| 1 | 0.54 |
| 2 | 0.47 |
| 3 | 0.40 |

Table 4: (*Real Data*) For the 65 ECs in the *general pool* where the model learned all the intended blindspots, we report average BDM DR with its standard error.

| Method | DR |
|---|---|
| Barlow | 0.09 (0.03) |
| Spotlight | 0.14 (0.04) |
| Domino | 0.38 (0.05) |
| **PlaneSpot** | **0.48** (0.05) |

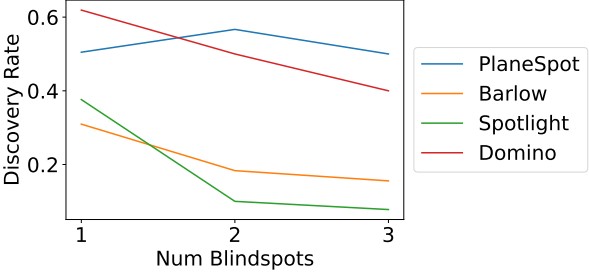

Figure 6: (*Real Data*) When we condition on only having less-specific induced blindspots, we see that BDM performance decreases as we increase the number of induced blindspots. These results are based on 82 ECs where the model learned all the intended blindspots.

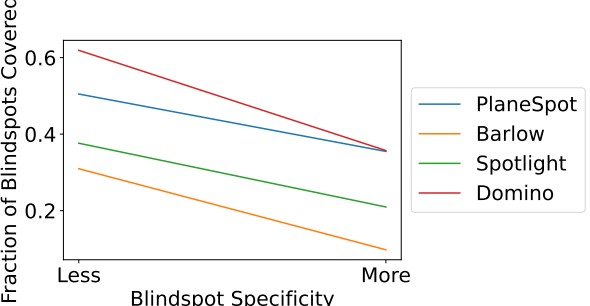

Figure 7: (*Real Data*) When we condition on only having a single induced blindspot, we see that BDMs are less effective at finding more-specific blindspots. These results are based on 70 ECs where the model learned the intended blindspots.

However, one key difference from our synthetic experiments is that when we use real data, we cannot guarantee that the model will learn the blindspots that we try to induce in it. Even if our goal is to train a model to underperform on all images with "elephants with people" by using a train set where all such images are mislabeled, the model may not actually have comparatively lower recall on images of "elephants with people" in the test set. We call this behavior "failing to learn" the intended blindspot. Further, if the model fails to learn the intended blindspots in a non-random way, the factors that we want to investigate may be correlated and, consequently, their effects may be confounded, which makes it difficult to estimate their individual effects. Table 3 shows that this confounding occurs between the effects of the number (Q2) and the specificity (Q3) of the blindspots in the COCO ECs.

To eliminate this confounding effect, we generate a new set of ECs for each factor where we hold the other confounding factor constant. We call the set of ECs where all of the factors are chosen independently the *general pool* because it covers a wider range of possible ECs, which is better for measuring overall BDM performance. We call this new set of ECs the *conditioned pool* because it is generated by conditioning the distribution used to generate the general pool on specific confounding factors. For all pools, we exclude all ECs where the model failed to learn the intended blindspots from our analyses. We provide details about the number of ECs retained in each pool in Appendix G. Because blindspot discovery is harder with real data, we use more lenient thresholds for our metrics and set $\lambda_p = \lambda_r = 0.5$.

**Results.** Overall, all of the findings that we observed using `SpotCheck` generalized to the COCO data experiments. In Table 4, we report the average BDM DR using ECs from the general pool. Because we only have knowledge of the blindspots that we induced, the DR is calculated relative to those blindspots and we cannot calculate BDM FDR. We observe that `PlaneSpot` has a DR competitive with existing BDMs on real data (Q1). Interestingly, Domino has a higher DR than Spotlight on the COCO data, which differs from the result observed in the synthetic data experiments in Table 2 where Spotlight had a higher DR than Domino. We explore this discrepancy further by performing ablations of the image representations used by Domino and Spotlight in Appendix I.

In Figures 6 and 7, we replicate the trends we observed using `SpotCheck`. Using the conditioned EC pools, we observe that BDM performance generally decreases as the number of induced blindspots increases (Q2) and that all BDMs are less effective at finding blindspots that are more-specific (Q3). Interestingly, we observe that Domino has a higher DR than `PlaneSpot` for models with a *single less-specific* induced blindspot (Figures 6 and 7). However, `PlaneSpot` has a higher DR than Domino for models with *multiple (less-specific)* induced blindspots (Figure 6), and a similar DR to Domino for models with a single *more-specific* blindspot (Figure 7. These differences explain why `PlaneSpot` has a higher DR than Domino for ECs in the general pool (Table 2), which includes models with multiple (more-specific) induced blindspots.

Finally, we ran an additional smaller-scale experiment detailed in Appendix H where we generated several ECs using data from the OpenImages dataset (Kuznetsova et al., 2018) to study the extent to which our observed trends hold on additional natural image benchmark datasets beyond MS-COCO. In summary, we found that `PlaneSpot` also achieves performance competitive with past BDMs on data from OpenImages.

## 6 Discussion

We provide an extended discussion of (1) similarities and differences between the synthetic and real data experiments (Section 6.1), (2) limitations of our evaluation and past evaluations of BDMs (Section 6.2), and (3) promising directions for future work on blindspot discovery (Section 6.3).

### 6.1 Interpreting the experimental results

In this work, we ran two sets of experiments to evaluate BDMs: experiments that use synthetic images generated from `SpotCheck`, and experiments that use real images from COCO. While we observe a few notable differences between their results, in general we find that *all three trends in BDM performance that we observed using* **SpotCheck** *were replicated on COCO* (a large image benchmark dataset).

One important observed difference is that even with more lenient thresholds for blindspot precision and recall, all BDMs have a much lower average discovery rate on real data (Table 4) than on synthetic data (Table 2). Thus, we caution researchers away from interpreting the raw values of metrics such as DR on `SpotCheck` as

an accurate representation of BDM DR on a more realistic or complex image dataset. Another observed difference is the relative performance of Domino and Spotlight. Our ablation experiments in Appendix I show that this performance difference is partly due to Domino's use of a CLIP embedding. We hypothesize that using CLIP is better suited to the COCO dataset, as COCO contains photographs that are likely more similar to CLIP's training dataset than synthetic images (Radford et al., 2021).

Despite these two differences, more broadly we observe that the majority of our experimental findings from `SpotCheck` also hold on real data. These results provide evidence that scientific findings from `SpotCheck` about the factors that influence BDM performance may generalize to a wider range of more realistic settings.

Our finding that several factors influence BDM performance has significant implications for BDM developers and users. As one example, we observed that *BDMs are less effective for models with a greater number of induced blindspots* in both the synthetic and real data experiments. This result is significant because past benchmarks such as Eyuboglu et al. (2022) only evaluate BDMs using models with a single induced blindspot, and may result in overly optimistic conclusions about BDM performance. Further, Li et al. (2022a;b) show that techniques that fix only a single blindspot often *worsen* model performance on other blindspots. This work is a cautionary example of how failing to discover an important blindspot can have negative consequences in practice, as practitioners who take action to address one blindspot may inadvertently worsen performance on other blindspots that they are unaware of.

## 6.2 Limitations

In this work, we propose `SpotCheck` as a new framework that addresses known limitations with prior approaches to evaluation by generating synthetic images. While synthetic images have many important benefits for benchmarking BDMs (*i.e.,* we can fully enumerate the set of semantic features that define each image dataset), ultimately we care about understanding and improving BDM performance in the real world. To understand if the trends we observed using `SpotCheck` generalize to settings with real (not synthetic) images, we also designed semi-controlled experiments using the COCO dataset. We observed that our findings did generalize, and that BDM performance generally deteriorates in more realistic and complex settings. Thus, we believe that characterizing and addressing existing BDM failure modes in a simple synthetic setting may serve as a stepping-stone to success in more realistic and complex settings.

While we believe that our contributions are an important step forward, we note that evaluating BDMs is difficult and several open challenges remain. One open question is whether the methodology that we used to induce blindspots in both our synthetic and real data experiments, *i.e.,* training with mislabeled images, influences BDM performance. In this work, we chose to *induce true blindspots* using available metadata so that we could run scientific experiments using models with different types of blindspots (*e.g.,* blindspots defined with different features or levels of specificity). While this strategy to induce blindspots is common in prior work (Eyuboglu et al., 2022; Kim et al., 2022), it remains an open question if blindspots that are induced in this way differ from naturally occurring blindspots.

Finally, we note that there are many open challenges associated with running semi-controlled experiments on real data that potentially bias or add noise to our real data results in Section 5.2. First, models trained on real data likely have naturally occurring blindspots (*i.e.,* blindspots that we did not intend the model to learn) that may influence the results. Second, the chosen BDM hyperparameters are probably sub-optimal because we cannot tune them for a specific EC. Third, the results may be biased in potentially unknown ways towards certain BDMs whenever some of the models in the set of ECs fail to learn the intended blindspots. For example, our general pool of ECs contains more less-specific blindspots than more-specific blindspots, which will bias the results towards BDMs that are more effective at finding less-specific blindspots. Fourth, there can be false positives in verifying that a model actually learned an intended blindspot. This is because there could be a sub-blindspot (that we may not have the metadata to define) within that intended blindspot that is actually causing the model to perform significantly worse on the intended blindspot. These challenges are shared by all existing quantitative evaluations of BDMs (Eyuboglu et al., 2022; Sohoni et al., 2020), and are precisely what motivated us to develop `SpotCheck`.

## 6.3 Future Work

**Addressing the discovered failure modes.** Our findings suggest several important challenges for future work that develops new BDMs to address. One such challenge is providing a practical way for practitioners

to tune BDM hyperparameters without access to knowledge of the true blindspots. Another challenge is to prevent BDMs from merging multiple true blindspots into a single hypothesized blindspot as observed in the additional experiments in Appendix F. A final challenge is to develop BDMs that can more effectively find more-specific (or "rare") blindspots.

**Evaluating BDMs in the wild.**   In this work, we evaluate BDMs by comparing the hypothesized blindspots they return to a set of true blindspots that we induce. While our proposed approach of inducing blindspots has several benefits, a remaining direction for future work is how to evaluate BDMs in more realistic settings where we have *no knowledge* of the model's true blindspots.

In Appendix K, we propose one potential approach to directly evaluate BDMs using a controlled user study where we show the hypothesized blindspots *to a human subject*, and ask them to write down text description hypotheses of groups where the model underperforms. For example, if a user hypothesizes the model underperforms on images of "zebras outside", we would evaluate the correctness of their hypothesis by calculating the model's performance on a new sample of images of "zebras outside". This approach avoids using knowledge of the true blindspots by simply evaluating the correctness of each individual hypothesis. Another benefit is that this approach can be used to evaluate the utility of a broader set of tools (*e.g.,* interactive interfaces or post-hoc explanations such as those listed in Section 2), unlike past evaluations which are restricted to fully-automated tools that return groups of datapoints.

**Interactive blindspot discovery.**   Our finding that `PlaneSpot`, a BDM that clusters on a simple 2D embedding, is competitive with state-of-the-art BDMs poses several promising directions for future work. Because a 2D embedding can be easily visualized, we can directly show the image representation to a practitioner. The practitioner can then directly discover blindspots from the image representation, effectively replacing the clustering algorithm ("Hypothesis Class") in Table 1.

As a minimal proof-of-concept, we discuss a simple case study where we qualitatively inspect the 2D scvis embedding for several large-scale benchmark datasets in Appendix J. We find that several true blindspots documented in past work *are* indeed separable in the 2D embedding space, but that many of them appear as a small cluster or other unusual shape that is unlikely to be returned by a standard clustering algorithm. We believe that human-in-the-loop clustering is promising as a human subject can iteratively refine the boundaries of each cluster and also has a richer contextual understanding of "coherence" to guide their clustering (Bae et al., 2020). While a growing number of interactive model debugging interfaces show users a 2D dataset embedding (Rajani et al., 2022; Cabrera et al., 2023; Suresh et al., 2023), its benefits relative to fully automated clustering have yet to be studied systematically.

## 7   Conclusion

In this work, we introduced `SpotCheck`, a synthetic evaluation framework for BDMs, and `PlaneSpot`, a simple BDM that uses a 2D image representation. Using `SpotCheck`, we identified several factors that influence BDM performance, including the number of induced blindspots, the specificity and features that define each blindspot, and BDM hyper-parameters. We then designed additional experiments that use the MS-COCO dataset to validate that trends we observed using `SpotCheck` also hold for models trained on a large image benchmark dataset. We found that `PlaneSpot` achieves performance competitive with existing BDMs on both synthetic and real data. Our findings suggest several promising directions for future work (Section 6.3) to address discovered failure modes of existing BDMs.

Beyond the insights made in the experiments that we ran, we believe that our evaluation workflow has laid further groundwork for a more rigorous science of blindspot discovery. Researchers interested in benchmarking their own BDM or answering scientific questions about blindspot discovery can first use `SpotCheck` to run controlled experiments where they have full knowledge of and control over the models' true blindspots. We hope that our experimental design for our semi-controlled real data experiments (*e.g.,* our discussion on how to control for confounding effects) can also serve as a guidepost for researchers interested in evaluating a new BDM or validating new trends observed using `SpotCheck` on real-world data. While evaluations that use real

data are harder and noisier (in part, because of the challenges that we identified in Sections 5.2 and 6.2), these experiments are also more realistic.

We note that our proposed evaluation workflow is particularly well suited for identifying factors that influence BDM performance. For researchers, understanding these factors is essential to running fair and informative experiments (*e.g.,* only evaluating BDMs in settings that only have one induced blindspot, which we have identified as being easier, may misrepresent how BDMs perform in more realistic settings where models have more than one true blindspot). For practitioners, understanding these factors may help them use any prior knowledge they may have to select an effective BDM for their own application domain. We release the `SpotCheck` source code used in our experiments as a public benchmark for BDMs (link) and provide an open-source implementation of `PlaneSpot` (link).

## 8 Acknowledgements

This work was supported in part by the National Science Foundation grants IIS1705121, IIS1838017, IIS2046613, IIS2112471, and funding from Meta, Morgan Stanley, Amazon, and Google. Any opinions, findings and conclusions or recommendations expressed in this material are those of the author(s) and do not necessarily reflect the views of any of these funding agencies.

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

# A `SpotCheck`, Extended

In this section we detail how we use `SpotCheck` to generate random experimental configurations.

- In Section A.1, we define the different types of semantic features that can appear in each image.
- In Section A.2, we define a synthetic image dataset, how we generate random datasets, and how we sample images from a dataset.
- In Section A.3, we define a blindspot for a synthetic image dataset, how we generate a random blindspot, and how we generate an unambiguous set of blindspots.

**Related Work.** `SpotCheck` is a synthetic evaluation framework for blindspot discovery that entails generating synthetic datasets and training models with known true blindspots. `SpotCheck` is inspired by (Kim et al., 2022), a benchmark for saliency maps, which aims to train models with known ground-truth reasoning. Our work is related to several other synthetic evaluation frameworks for shortcut learning, including Eulig et al. (2021); Hermann & Lampinen (2020). Our work differs from these other related works in that we specifically aim to induce different true blindspots in each model.

## A.1 Semantic Features

Table 5 defines all of the semantic features that `SpotCheck` uses to generate synthetic images. We call these semantic features *Attributes* and group them into *Layers* based on what part of an image they describe. Each Attribute has two possible *Values*: a Default and Alternative Value. Each synthetic image has an associated list of (Layer, Attribute, Value) triplets that describes the image. Figure 8 shows this triplet list for two synthetic images.

We sometimes refer to the Square/Rectangle/Circle/Text Layers as *Object Layers* because they all describe a specific object that can be present in an image. The location of each object within an image is chosen randomly, subject to the constraint that each object doesn't overlap with any other object.

Table 5: The Layers and Attributes that define the synthetic images.

| Layer | Attribute | Default Value | Alternative Value |
|---|---|---|---|
| Background | Color | White | Grey |
| | Texture | Solid | Salt and Pepper Noise |
| Square/Rectangle/Circle/Text | Presence | False | True |
| | Size | Normal | Small |
| | Color | Blue | Orange |
| | Texture | Solid | Vertical Stripes |
| Square (continued) | Number | 1 | 2 |

### A.2 Defining a Dataset using these Semantic Features

At a high level, `SpotCheck` defines a *Dataset* by deciding whether or not each Attribute of each Layer is *Rollable* (*i.e.,* the Attribute can take either its Default or Alternative Value, uniformly at random) or not Rollable (*i.e.,* the Attribute only takes its Default Value). We measure a Dataset's complexity using the number of Rollable Attributes it has. Figure 8 describes the Rollable and Not Rollable Attributes for an example Dataset.

**Generating a Random Dataset.** We start by picking which Layers will be part of the Dataset:

- Images need a background, so all Datasets have the Background Layer.
- The task is to predict whether there is a square in the image, so all Datasets have the Square Layer.
- We add 1-3 (chosen uniformly at random) of the other Object Layers (chosen uniformly at random without replacement from the set {Rectangle, Circle, Text}) to the Dataset.

Once the Layers are chosen, we make 6-8 (chosen uniformly at random) of the Attributes Rollable:

- Each Object Layer has its Presence Attribute made Rollable.
- Then, the remaining Rollable Attributes are chosen by iteratively:
  - Selecting a Layer uniformly at random from those that have at least one Not Rollable Attribute.
  - Selecting an Attribute from that Layer uniformly at random from those that are Not Rollable.

**Sampling an Image from a Dataset.** Once a Dataset's Rollable Attributes have been defined, generating a random image is straightforward:

- For each Attribute from each Layer in the Dataset, we pick a random Value if the Attribute is Rollable. Attributes that are Not Rollable will take their Default Value.
  - If the Layer is an Object Layer:
    - If the Presence Attribute is True, the location of the object is chosen randomly (subject to the non-overlapping constraint).
    - If the Presence Attribute is False, the object will not be rendered (regardless of the Values chosen for the other Attributes of this Layer).
- We then use the resulting (Layer, Attribute, Value) triplet list and the list of object locations to render a 224x224 RGB image.
- Finally, we calculate any MetaAttributes (explained next) and append these (Layer, MetaAttribute, Value) triplets to the image's definition list.

**Calculating MetaAttributes.** While each Attribute corresponds to a semantic feature, there are a potentially infinite number of MetaAttributes that one could calculate as semantically meaningful functions of an image. We list the MetaAttributes that we calculate in our experiments in Table 6. Because this space is infinitely large and grows with the number of Attributes, we exclude MetaAttributes from our measure of Dataset complexity.

**Dataset Definition:**
Rollable Attributes

```
Background: {Color: False,
              Texture: True}

Square: {Presence: True,
         Size: False,
         Color: True,
         Texture: True,
         Number: False}

Text: {Presence: True,
       Size: False,
       Color: True,
       Texture: False}
```

Example Image #1

```
(Background, Texture, Solid),

(Square, Presence, True),
(Square, Color, Blue),
(Square, Texture, Vertical
Stripes),

(Text, Presence, True),
(Text, Color, Orange}
```

Example Image #2

```
(Background, Texture, Salt &
Pepper),

(Square, Presence, False),
(Square, Color, Blue),
(Square, Texture, Solid),

(Text, Presence, True),
(Text, Color, Blue)
```

Figure 8: **Top Row.** The definition of an example Dataset generated by `SpotCheck`. Notice that this Dataset has 3 Layers and 6 Rollable Attributes. **Middle/Bottom Row.** Two example images generated from this Dataset along with their (Layer, Attribute, Value) triplet lists. Notice that Not Rollable Attributes in this Dataset take on their Default Values in these images and are not in the images' triplet lists.

Table 6: The MetaAttributes that we calculate for each synthetic image.

| Layer | MetaAttribute | Value | Meaning |
|---|---|---|---|
| Background | Relative Position | 1 | Square is above the horizontal centerline of the image |
| | | 0 | Square is bellow the horizontal centerline of the image |
| | | -1 | No Square |

### A.3 Defining the Blindspots for a Dataset

`SpotCheck` defines a *Blindspot* using a list of (Layer, (Meta)Attribute, Value) triplets. We measure a Blindspot's specificity using the length of its definition list. An image belongs to a blindspot if and only if the Blindspot's definition list is a subset of the image's definition list. Figure 9 shows two example Blindspots.

**Generating a Random Blindspot.** `SpotCheck` generates a random Blindspot consisting of 5-7 (chosen uniformly at random) (Layer, (Meta)Attribute, Value) triplets for a Dataset by iteratively:

- Selecting a Layer (uniformly at random from those that have at least one Rollable Attribute[3] that is not already in this Blindspot)
- Selecting a Rollable Attribute from that Layer:
  - Object Layers: If the Layer's Presence Attribute is not in this Blindspot, select its Presence Attribute. Otherwise, select an Attribute uniformly at random from those that are not already in this Blindspot and set the Layer's Presence Attribute Value to True for this Blindspot.
  - Background Layers: Select an Attribute uniformly at random from those that are not already in this Blindspot.
- Selecting a Value for that Attribute (uniformly at random)

Notice that, if an Object Layer is selected more than once, then we ensure that the Object's Presence Attribute has a Value of True in the Blindspot definition. We enforce this *Feasibility Constraint* to ensure that every triplet in the Blindspot's definition list correctly describes the images belonging to the Blindspot (*e.g.,* [(Circle, Presence, False), (Circle, Color, Blue)] is infeasible because an image with a blue circle must have a circle in it).

**Generating an Unambiguous Set of Blindspots.** For each Dataset, we generate 1-3 (chosen uniformly at random) Blindspots using the process described above. However, when generating multiple blindspots, they can be *ambiguous* which causes problems when using them to evaluate BDMs.

***Definition.*** A set of Blindspots, $S_1$, is ambiguous if there exists a different set of Blindspots, $S_2$, such that both:

1. The union of images belonging to $S_1$ is equivalent to the union of images belonging to $S_2$. As a result, $S_1$ and $S_2$ would both correctly describe the model's blindspots.
2. An evaluation that uses Discovery Rate (Equation 6) would penalize a BDM if it returns $S_2$ instead of $S_1$. More precisely, $\mathrm{DR}(S_2, S_1) < 1$ for $\lambda_p = \lambda_r = 1$.

***Example.*** Suppose that we have a very simple Dataset with two Rollable Attributes, $X$ and $Y$ which are uniformly distributed and independent, and consider two different sets of Blindspots for this Dataset:

- $S_1 = \{B_1, B_2\}$ where $B_1 = [(X = 1)]$ and $B_2 = [(X = 0), (Y = 1)]$
- $S_2 = \{B_1', B_2'\}$ where $B_1' = [(X = 1), (Y = 0)]$ and $B_2' = [(Y = 1)]$

Then, $S_1$ is ambiguous because:

- $S_1$ and $S_2$ induce the same behavior in the model: they both mislabel an image if $X = 1 \vee Y = 1$.
- A BDM would be penalized for returning $S_2$:

$$\mathrm{BP}(B_1', B_1) = 1.0 \wedge \mathrm{BP}(B_1', B_2) = 0 \wedge \mathrm{BP}(B_2', B_1) = \mathrm{BP}(B_2', B_2) = 0.5 \implies$$

$$\mathrm{BR}(S_2, B_1) = 0.5 \wedge \mathrm{BR}(S_2, B_2) = 0 \implies$$

$$\mathrm{DR}(S_2, S_1) = 0$$

In fact, for this example, there are only two sets of two unambiguous Blindspots, ($\{[(X = 0), (Y = 0)], [(X = 1), (Y = 1)]\}$ and $\{[(X = 0), (Y = 1)], [(X = 1), (Y = 0)]\}$), and there exists no unambiguous set of three Blindspots.

***Preventing Ambiguity.*** In general, ambiguity occurs whenever the union of two blindspots forms a contiguous region in the discrete space defined by the Rollable Attributes. Consequently, we prevent

---

[3]All MetaAttributes are considered to be "Rollable" when generating a random Blindspot.

ambiguity by ensuring that any pair of blindspots has at least two of the *same* Rollable Attributes with *different* Values in their definition lists. We call this the *Ambiguity Constraint.*

***Implications of the Ambiguity and Feasibility Constraints.*** In our experiments, our goal is to generate experimental configurations with a diverse set of Datasets and associated Blindspots. However, the Ambiguity Constraint (AC) and Feasibility Constraint (FC) limit the number of valid Blindspots for any specific Dataset.

To see this, notice that the AC places more constraints on each successive Blindspot added to an experimental configuration. This has two implications. First, that generating an experimental configuration with more Blindspots requires a Dataset with more Rollable Attributes (more complexity) and Blindspots with more triplets (more specificity). Further, because we cannot set the Attribute Values of a Blindspot's triplets independently of each other [FC], we need more complexity and specificty than a simple analysis based only on the AC suggests. Second, that each successive Blindspot is more closely related to the previous ones which means that larger sets of Blindspots are "less diverse" or "less random" in some sense.

With these trade-offs in mind, we generated experimental configurations with:

- Background, Square, and 1-3 other Object Layers
- A total of 6-8 Rollable Attributes
- 1-3 Blindspots
- 5-7 triplets per Blindspot

because an experimental configuration with any combination of these values is able to satisfy the AC and the FC while still having a diverse set of Blindspots.

| **Blindspot #1** | **Blindspot #2** |
|---|---|
| (Background, Texture, Salt & Pepper), | (Background, Texture, Salt & Pepper), |
| (Square, Presence, True), | (Square, Presence, True), |
| (Square, Color, Orange), | (Square, Color, Orange), |
| (Square, Texture, Striped), | (Square, Texture, Striped), |
| (Text, Presence, True) | (Text, Presence, True), |
| | (Text, Color, Orange) |
| "images with salt & pepper backgrounds, blue striped squares, and text" | "images with salt & pepper backgrounds, blue striped squares, and orange text" |

**Example Images**

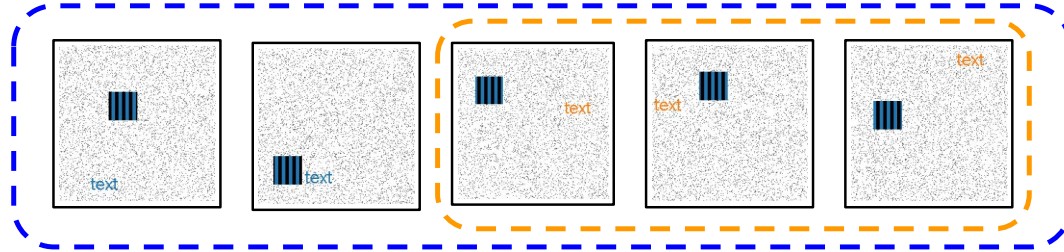

Figure 9: Two example Blindspots, Blindspot #1 (Left) and Blindspot # 2 (Right) for the example Dataset from Figure 8. Each Blindspot is defined by a list of (Layer, (Meta)Attribute, Value) triplets as shown above. We also display example images belonging to each Blindspot: all of the images inside of the blue border belong to Blindspot #1, while only the subset of images inside of the orange border belong to Blindspot #2. In this example, Blindspot #2 is more specific (defined using 6 semantic features) than Blindspot #1 (defined using 5 semantic features).

# B   Ablation: Method to Induce Blindspots

In this section, we present the results of additional experiments where we vary the method used to induce true blindspots in several `SpotCheck` ECs. We also further justify why we chose to train using mislabeled images, and possible limitations of this approach.

**Why train with mislabeled images?**   In our experiments, we induce true blindspots by fine-tuning a pretrained model using a dataset where all images that belong to a blindspot are mislabeled. This method (training with mislabeled images to induce true blindspots) is shared by several prior studies, including Domino (the other existing quantitative evaluation framework for BDMs) (Eyuboglu et al., 2022) and Kim et al. (2022).

In Section 6.2, we note that the method that we used to induce blindspots may not be representative of how blindspots emerge in practice (i.e. it is unlikely that all of the images in each true blindspot were mislabeled during training). However, after trying multiple alternatives, we ultimately chose this method because we found it to be the most reliable way to induce the desired blindspots in a fully synthetic setting. We describe these experiments below.

**Additional experiments.**   We ran a set of additional experiments where we varied the method used to induce the true blindspots for 50 held-out `SpotCheck` ECs. We present results for two alternative methods: training with lower label noise rates and excluding the true blindspot from training.

**Method #1: Varying the label noise rate**   We also ran an ablation where instead of training on a dataset where all of the images belonging to a blindspot were mislabeled (i.e. 100% label noise), we instead randomly flipped each label with probability $\alpha$. We tried two different label noise percentages: 50% and 30% (the largest label noise rate used by Eyuboglu et al. (2022)). We observed that training with label noise rate $\alpha$ results in succeeding to learn the intended blindspots for 36 out of 50 ECs (72%) for noise rate 0.5, and 5 out of 50 ECs (10%) for noise rate 0.3.

**Method #2: Excluding each true blindspot from training**   We ran an ablation where we held out all images belonging to a true blindspot during training. Unfortunately, we observed that this method results in failing to learn the intended blindspots for all 50 ECs: i.e. when we exclude the blindspot from the train set, the model does not have significantly worse recall on images belonging to each blindspot in the test set.

In summary, we observed that none of the alternative methods that we tried were as successful at inducing all of the true blindspots. In contrast, training with mislabeled images had a 100% success rate at inducing the true blindspots. As discussed in Section 5.2, failing to learn all of the intended blindspots can cause undesirable confounding effects that make it difficult to study the experimental factors that influence blindspot discovery.

## C   How do our definitions of "belonging to" and "covering" build on similar ideas in existing work?

At a high level, we refine these ideas so that they better reflect the fact that, in practice, a user is going to look at the hypothesized blindspots returned by a BDM in order to come to conclusions about the model's true blindspots. This entails making three specific changes.

First, our definition of "belonging to" uses a high, in absolute terms, threshold for $\lambda_p$ (*i.e.,* 0.8) in order to minimize the amount of noise the user has to deal with when determining the semantic definition of a blindspot. This is in contrast to a relative definition (*e.g.,* better than random chance) which would lead to significant amounts of noise for uncommon blindspots. For example, Table 2 from Sohoni et al. (2020) shows that as few as one in seven images in the hypothesized blindspots are actually from the true blindspots for the Waterbirds and CelebA datasets. While better than random chance, the output of the BDM is still very noisy, which is likely to create problems for the user.

Second, our definition of "covering" incorporates blindspot recall in order to prevent the user from coming to conclusions that are too narrow. For example, consider the middle row of Figure 5 from Eyuboglu et al. (2022). In these examples, the blindspots are defined as "people wearing glasses," "people with brown hair," or "smiling people." However, the BDM returns images that are of "men wearing glasses," "women with brown hair, and "smiling women." As a result, the user may incorrectly conclude that the model is exhibiting gender bias because the BDM led them to conclusions that are too narrow.

Third, our definition of "covering" allows for hypothesized blindspots to be combined in order to take advantage of the user's ability to synthesize what they see and come to more general conclusions, which is something that would be very challenging to do algorithmically. Experimentally, we observe that this combining is essential for some of these BDMs to work on `SpotCheck`. Further, in Appendix J, we qualitatively observe that this useful for real problems as well. For example, the "ties without people" blindspots includes a set of images of "cats wearing ties" that are separate, in the model's representation space, from the other images in the blindspot and that, as a result, are likely to be returned separately by a BDM.

## D  `PlaneSpot`: **Ablations**

We run additional experiments where we vary the dimensionality reduction method used by `PlaneSpot`, using both synthetic and real data:

| Method | $d$ | DR | DR SE | FDR | FDR SE |
|--------|-----|-----|-------|------|--------|
| scvis | $d = 2$ | 0.85 | 0.045 | 0.027 | 0.016 |
| scvis | $d = 4$ | 0.86 | 0.043 | 0.021 | 0.015 |
| scvis | $d = 8$ | 0.66 | 0.059 | 0.029 | 0.017 |
| scvis | $d = 16$ | 0.53 | 0.066 | 0.016 | 0.016 |
| PCA | $d = 2$ | 0.70 | 0.062 | 0.026 | 0.018 |
| tSNE | $d = 2$ | 0.86 | 0.04 | 0.011 | 0.011 |

Table 7: (*Synthetic Data*) The average DR and FDR (and their standard errors, "SE") on a hold-out set of 50 `SpotCheck` ECs when we run modified versions of `PlaneSpot` that use different dimensionality reduction methods (with varying output dimension $d$).

| Method | $d$ | DR | DR SE |
|--------|-----|-----|-------|
| scvis | $d = 2$ | 0.48 | 0.047 |
| PCA | $d = 2$ | 0.33 | 0.047 |
| tSNE | $d = 2$ | 0.48 | 0.048 |

Table 8: (*Real Data*) The average DR and its standard error "SE" on the *general pool* of COCO ECs, when we run modified versions of `PlaneSpot` that use different dimensionality reduction methods.

In summary, we find that scvis and tSNE (van der Maaten & Hinton, 2008) achieve a better DR and FDR than PCA on both synthetic and real data. In our synthetic data experiments (Table 7), we observe that scvis's performance does not degrade as we decrease the output dimension $d$. Finally, we observe scvis and tSNE have comparable performance on both synthetic and real data. We ultimately chose to use scvis rather than tSNE for its relative computational efficiency on large datasets (Ding et al., 2018).

# E   Hyperparameter Tuning

To tune hyperparameters, we use a secondary set of 20 ECs to calculate BDM DR and FDR. While some of these BDMs have multiple hyperparameters, we focused on the main hyperparameter for each BDM and left the others at their default values.

**Barlow (Singla et al., 2021).** The main hyperparameter is the *maximum depth* of the decision tree whose leaves define the hypothesized blindspots. The results of the hyperparameter search are in Table 9.

**Spotlight (d'Eon et al., 2021).** The main hyperparameter is the *minimum weight* assigned to a hypothesized blindspot. The results of the hyperparameter search are in Table 10.

**Domino (Eyuboglu et al., 2022).** The main hyperparmeter is $\gamma$ which controls the relative importance of coherence and under-performance in the hypothesized blindspots. The results of the hyperparameter search are in Table 11.

**PlaneSpot.** For our BDM, the main hyperparameter is $w$ which controls the *weight* of the model-confidence dimension relative to the two spatial dimensions from the scvis embedding. The results of the hyperparameter search are in Table 12.

Table 10:   The hyperparameter tuning results for Spotlight. We use a minimum weight of 0.01 in our experiments. In Figure 2 we show that, despite the fact that 0.01 and 0.02 have very similar average results, they behave very differently.

| Minimum Weight | DR | FDR |
|---|---|---|
| 0.005 | 0.49 | 0.04 |
| 0.01 | 0.88 | 0.06 |
| 0.02 | 0.88 | 0.08 |
| 0.04 | 0.48 | 0.00 |

Table 9:   The hyperparameter tuning results for Barlow.  For maximum depths of 11 and 12, we noticed that Barlow's outputs matched exactly for all of the configurations. This indicates that it never learned a decision tree of depth 12. As a result, we use a maximum depth of 11 for our experiments.

| Maximum Depth | DR | FDR |
|---|---|---|
| 3 | 0.07 | 0.0 |
| 4 | 0.14 | 0.0 |
| 5 | 0.19 | 0.0 |
| 6 | 0.27 | 0.12 |
| 7 | 0.36 | 0.15 |
| 8 | 0.37 | 0.09 |
| 9 | 0.43 | 0.11 |
| 10 | 0.43 | 0.08 |
| 11 | 0.48 | 0.07 |
| 12 | 0.48 | 0.07 |

Table 11:   The hyperparameter tuning results for Domino. We use $\gamma = 10$ in our experiments.

| $\gamma$ | DR | FDR |
|---|---|---|
| 5 | 0.48 | 0.06 |
| 10 | 0.72 | 0.03 |
| 15 | 0.70 | 0.06 |
| 20 | 0.60 | 0.09 |

Table 12:   The hyperparameter tuning results for PlaneSpot. We use $w = 0.025$ in our experiments.

| $w$ | DR | FDR |
|---|---|---|
| 0 | 0.57 | 0.00 |
| 0.025 | 0.95 | 0.00 |
| 0.05 | 0.88 | 0.00 |
| 0.1 | 0.87 | 0.00 |

# F  Why are these BDMs failing?

In this section, we are going to try to identify, from a methodological standpoint, why these BDMs are failing. We start by defining some additional metrics that allow us to understand why a BDM failed to cover a specific true blindspot. Then, we describe how we average those metrics across our ECs to observe general patterns. Finally, we give some possible explanations for the observed trends.

**Explaining a Specific Failure.** Given the output of a BDM, $\hat{\boldsymbol{\Psi}}$, we want to understand why it failed to cover a specific true blindspot, $\Psi_m$. To do this, we measure the fraction of the images in that true blindspot, $i \in \Psi_m$, that fall into each of four categories:[4]

- $i$ was *not returned* by the BDM. Intuitively, this tells us how often the BDM does not show an image to the user that it should. Specifically, this means that $i \notin \hat{\Psi}_k \; \forall k$.
- $i$ is *found* by the BDM. Intuitively, this tells us how close the BDM is to covering $\Psi_m$. Specifically, this means that $\exists k \mid i \in \hat{\Psi}_k \wedge \mathrm{BP}(\hat{\Psi}_k, \Psi_m) \geq \lambda_p$.
- $i$ was part of a *merged* blindspot according to the BDM. Intuitively, this tells us how often the BDM merges multiple true blindspots into a single hypothesized blindspot. Specifically, this means that $\exists k \mid i \in \hat{\Psi}_k \wedge \mathrm{BP}(\hat{\Psi}_k, \cup_m \Psi_m) \geq \lambda_p$. Note that $\mathrm{BP}(\hat{\Psi}_k, \cup_m \Psi_m)$ measures the fraction of the images in $\hat{\Psi}_k$ that belong to *any* true blindspot.
- $i$ was part of a *impure* blindspot according to the BDM. Intuitively, this tells us how often the BDM includes too many images that do not belong to any true blindspot into a hypothesized blindspot. Specifically, this means that $\mathrm{BP}(\hat{\Psi}_k, \cup_m \Psi_m) < \lambda_p \; (\forall k \mid i \in \hat{\Psi}_k)$.

**Averaging to gain more general patterns.** While these metrics can help us understand why a BDM failed to cover a specific true blindspot from a specific EC, we want to arrive at more general results. To do this, we first average these metrics across the true blindspots in an EC that are not covered by the BDM, $\{\Psi_m \mid \mathrm{BR}(\hat{\boldsymbol{\Psi}}, \Psi_m) < \lambda_r\}$, and then average them across the ECs where the BDM did not cover all of the true blindspots, $\{\boldsymbol{\Psi} \mid \mathrm{DR}(\hat{\boldsymbol{\Psi}}, \boldsymbol{\Psi}) \neq 1\}$. Table 13 shows the results.

**Explaining observed trends.** Our main observation is that a significant portion of all of the BDMs' failures can be explained by the fact that they tend to merge multiple true blindspots into a single hypothesized blindspot. However, this raises the question: is the a failure of the BDMs or are they simply inheriting the problem from their image representation, which is failing to separate the true blindspots? To address this question, we manually inspected the 2D scvis embedding used by `PlaneSpot`, which exhibits this failure the most strongly, for the 10 ECs where at least 50% of the images in the true blindspots that were not covered belonged to a hypothesized blindspot that merges multiple true blindspots. For 8 of those 10 ECs, the true blindspots were easily visually separable in this 2D embedding, which means that the true blindspots are also separable in the original image representation. Consequently, we conclude that this is a failing of the BDMs.

Additionally, we make two less significant observations. First, that Spotlight is the only BDM that fails to return a non-trivial fraction of the images that it is should. Because it is the only BDM that does not do this, we suggest that BDMs should partition the entire input space (including partitions that do not have higher than average error). Second, that Barlow is the only that has more failures due to returning impure hypothesized blindspots than merged blindspots. Between this and Barlow's generally poor performance, this suggests that axis-aligned decision trees are not a particularly promising hypothesis class for BDMs.

Table 13: The average value of each of the four metrics we use to explain why a BDM failed to cover a true blindspot.

| Method | Not Returned | Found | Merged | Impure |
|---|---|---|---|---|
| Barlow | 0.02 | 0.15 | 0.39 | 0.44 |
| Spotlight | 0.13 | 0.46 | 0.33 | 0.04 |
| Domino | 0.0 | 0.38 | 0.36 | 0.24 |
| PlaneSpot | 0.0 | 0.0 | 0.57 | 0.42 |

---

[4]Every image, $i$, falls into exactly one of these categories. The definition of each category is written under the assumption that $i$ does not fall into any of the previous categories.

# G   COCO Experiments, Details

We detail the methodology used to generate ECs using the COCO dataset (Lin et al., 2014).

## G.1   Prediction Task & Data Preprocessing

COCO is a large-scale object detection dataset. We use the 2017 version of the dataset. We re-sample from the given train-test-validation splits to have a larger test set of images for use in blindspot discovery. Our final train, test, and validation sets have 66874, 44584, and 11829 images respectively.

**Prediction Task.** We define the binary classification task as detecting whether an object belonging to some *super-category* (containing multiple of the 80 object categories) is present in the image. In our experiments, each EC is associated with 1 of 3 different super-category prediction tasks: detecting if an *animal* (*e.g.,* a dog, cat, etc), a *vehicle* (*e.g.,* a car, airplane, etc), or a *furniture item* (*e.g.,* a chair, bed, etc) are present in an image. We use the super-category definitions provided in the COCO paper (Lin et al., 2014). We sample ECs from multiple (as opposed to only a single) prediction tasks so that we have a larger number of possible blindspot definitions.

**EC Preprocessing.** For each EC, we use the EC's prediction task to down-sample the original COCO dataset to ensure that the dataset's blindspots are identifiable (*i.e.,* so that there is no ambiguity or overlap in the blindspots we induce in our models). Specifically, we drop all images that have more than 1 unique object category belonging to the task super-category (*e.g.,* if the task is to detect animals, we drop all images that have both a dog and a cat). We also down-sample so that the final train, test, and validation sets are all class-balanced.

## G.2   Blindspot Definitions

All blindspots in our ECs are defined using an object category that is a subset of those belonging to the task super-category. For example, an EC with the "animal" prediction task can have the blindspot "zebra", as the zebra object category belongs to the animal super-category (Figure 5).

**Specificity.** To measure the effect of blindspot specificity, we generate two different *types* of blindspots: more and less specific. Less-specific blindspots are defined using only one object category, such as the "zebra" blindspot. More-specific blindspots are defined using two object categories. Like the less-specific blindspots, one of the object categories must belong to the task super-category (*e.g.,,* must be an animal for an animal prediction EC). But, the second object category used to define a more-specific blindspot must *not* belong to the task super-category. For example, the second object can be "person" (but not "dog") because a person is not an animal.

Each more-specific blindspot is then defined using (1) the presence of the object belonging to the super-category, and (2) either the presence *or absence* of the object outside of the super-category. For example, valid blindspot definitions include "images of elephants with people" (task: animal), "images of bicycles without a backpack" (task: vehicle), or "images of dining tables with forks" (task: furniture item). We chose to allow some blindspots to be defined using the *absence* of a second object inspired by the past observation that models can incorrectly rely on the presence of spurious artifact objects instead of identifying the true object class (Plumb et al., 2022), consequently underperforming on images without these artifacts.

## G.3   Model Training

Similarly to our experiments using `SpotCheck`, we induce true blindspots by training with the wrong labels for images belonging to blindspots. For each EC, we train a Resnet-18. We perform model selection by picking the model with the best validation set loss.

**Validating that models learn the induced blindspots.** Because real data is more complex than synthetic data, it is possible that even when we train with the wrong labels for a particular subgroup, the model may not underperform on other images belonging to the subgroup. We call this behavior *failing to induce* a true blindspot. For each true blindspot, we measure if the blindspot was successfully induced by comparing the

model's performance on images belonging to, vs. outside of the blindspot in the separate *test set*. We only retain ECs where the model has at least a 20% recall gap inside vs. outside of the blindspot.

## G.4 ECs: General Pool

We began our analysis by replicating the procedure we used to sample random ECs using `SpotCheck` as closely as possible. Recall that when using `SpotCheck` to measure the effect of various factors, we generated a pool of ECs where each factor (*e.g.,*, the number of model blindspots and each blindspot's specificty) were chosen *independently*. We repeat this general procedure using the COCO dataset, and call these ECs the "General Pool".

We provide pseudocode that we use to sample a pool of $N = 90$ ECs:

- For each super-category **prediction task** in {animal, vehicle, furniture item }:
  - For each **number of blindspots** in $[1, 2, 3]$:
    - Generate 10 experimental configurations. For each configuration:
      - For each blindspot, choose the blindspot's definition:
        - Choose the blindspot's **specificity** uniformly at random from {More-Specific, Less-Specific}.
        - Choose an **object category** that belongs to the task super-category uniformly at random from those that were not already used in the EC's other blindspots.
        - If the blindspot is **more-specific**:
          - Choose a second **object category** that does not belong to the task super-category uniformly at random from those that are *eligible*. An object category is eligible if it has not yet been used to define another of the ECs blindspots, and if both the intersection and set difference of the first and second object categories have at least 100 images in the train set. We impose these eligibility constraints to avoid selecting blindspots that are too small, as we hypothesize that larger blindspots may be easier to induce.
          - Choose if the blindspot will be defined using the {Presence, Absence} of the second object uniformly at random.

Note that for each EC sampled using the above pseudocode, the EC's number of blindspots and each indivudal blindspot's specificity are selected independently.

Table 14: Counts the retained ECs in the general pool (where all blindspots are successfully induced) for each prediction task and number of blindspots.

| Prediction Task | Number of blindspots | **Count** of (ECs retained for analysis) / (total ECs sampled) |
|---|---|---|
| **animal** | 1 | 10 / 10 |
| | 2 | 8 / 10 |
| | 3 | 7 / 10 |
| **furniture** | 1 | 8 / 10 |
| | 2 | 6 / 10 |
| | 3 | 5 / 10 |
| **vehicle** | 1 | 8 / 10 |
| | 2 | 6 / 10 |
| | 3 | 7 / 10 |

**Summary Statistics.** Of the 90 sampled ECs in the general pool, we retain 65 for our analysis, dropping 25 where we failed to induce the sampled blindspot. Table 14 summarizes the prediction tasks and number of blindspots in the retained general pool of ECs.

## G.5 ECs: Conditioned Pools

**Number of Blindspots: Conditioned Pool.** We provide pseudocode that we use to sample a pool of $N = 84$ ECs. We hold each blindspot's specificity constant and only sample less-specific blindspots.

Before sampling the conditioned pool ECs, we first train several models where for each model, we try to induce a single COCO object category as a blindspot. We sample the blindspots for the conditioned distribution only from the subset of $C$ object categories that were successfully induced as singleton blindspots.

- For each super-category **prediction task** in {animal, vehicle, furniture item }:
  - For each **number of blindspots** in $[1, 2, 3]$:
    - Using the $C$ object categories that belong to the task super-category *and* that were able to be learned as singleton blindspots, sample up to $N = 20$ ECs from the list of $C$ nCr (number of blindspots) possible combinations.

We only have 22 ECs with 1 (singleton) blindspot because there are only 22 total object categories that can be induced as singleton blindspots. All blindspots were successfully induced for the 84 sampled ECs. To measure the effect of the number of blindspots, we compare BDM performance the $22, 30$, and $30$ ECs with $1$, $2$, and $3$ blindspots respectively.

**Blindspot Specificity: Conditioned Pool.** We provide pseudocode that we use to sample a pool of $N = 70$ ECs. We hold the number of blindspots constant and only sample ECs with a single blindspot.

- For each super-category **prediction task** in {animal, vehicle, furniture item }:
  - Using the $C$ object categories that belong to the task super-category *and* that were able to be learned as singleton blindspots, construct a list of all of the *eligible* blindspots:
    - For the less-specific blindspots, all $C$ sub-categories are eligible.
    - For the more-specific blindspots, we define eligibility equivalently to the general pool: *i.e.,* for each $C$ object categories belonging to the task super-category, a second object category (that does not belong to the task super-category) is a *eligible* if both the intersection and set difference of the first and second object categories have at least 100 images in the train set. Concretely, the list of eligible blindspots consists of all eligible pairs of objects with both options of whether the blindspot is defined using the {Presence, Absence} of the second object.
    - From the list of eligible blindspots, sample up to 20 ECs (each with 1 blindspot) without replacement.

Of the 60 generated more-specific ECs, we retain 48 ECs for analysis, dropping the 12 ECs where we failed to induce the sampled blindspot. Like the other conditioned pool, there are only 22 possible less-specific ECs. To measure the effect of blindspot specificity, we compare BDM performance on the 48 more-specific to the 24 less-specific ECs.

# H   OpenImages Experiments

To validate if the trends we observed generalize to an additional large benchmark dataset, we designed additional experiments using data from OpenImages (Kuznetsova et al., 2018). Like MS-COCO, OpenImages is a large-scale object detection dataset of photographs that aims to depict objects in their natural contexts. Because OpenImages is a multi-classification task, we have access to *metadata* about the objects present in each image that we can use to define blindspots. We detail the methodology used to generate ECs below.

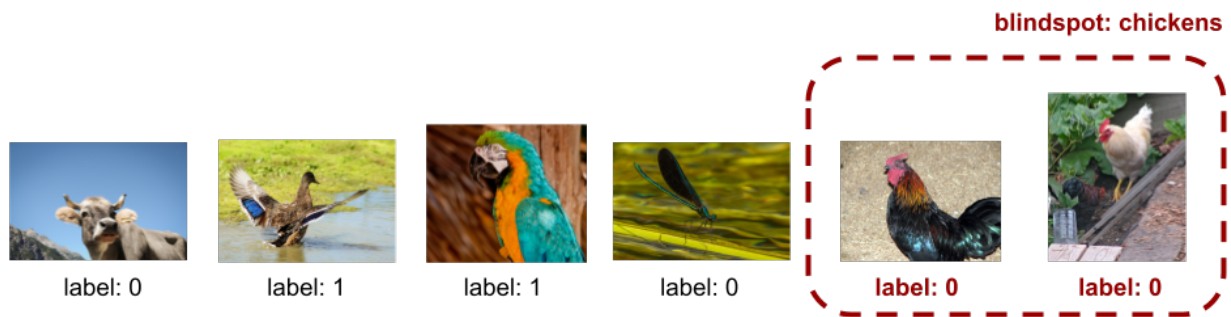

Figure 10: Images and their labels sampled from an example EC from our OpenImages experiments. Images where all of the pictured animals are birds are given a positive label, vs. images of animals that are not birds have a negative label. In this example EC, the blindspot is "chickens", a type of bird, which are mislabeled in the train set.

## H.1   Predcition Task & Preprocessing

We use data from the 1.9M subset of OpenImages V7 which has labels that have been verified by human annotators. This version of the dataset has 600 human-annotated classes, which are arranged in a hierarchy. We use a subsample of the complete OpenImages dataset where an *animal* is present in the object. We define the binary prediction task as detecting the animals' species, *i.e.,* whether the pictured animal(s) are birds. We drop all images that have two or more animals belonging to different species, and label all images that have an animal that is a child of the "bird" class as positive, and all other images of animals as negative. Finally, we down-sample until we've achieved class balance. Our final dataset contained 8000 positive images of birds, and 8000 negative images of other animals (that are not birds, such as mammals or invertebrates). We show example images and their labels in Figure 10. We also defined a custom 50-40-10 train-test-val split so that we had sufficient data to discover blindspots on the test set.

## H.2   Blindspot Definitions

In our experiments, we generated a *conditioned pool* of blindspot definitions that were all defined using a single object class (*i.e.,* are "less specific" blindspots using our terminology from Section 5.2). Specifically, each blindspot is defined as one of the 9 possible birds (children of the "bird" class) where we had at least 1000 images in the dataset: "chicken", "eagle", "duck", "goose", "swan", "falcon", "parrot", and "sparrow". Like our COCO experiments, we exclude all images that have more than 1 unique type of bird (*e.g.,* we would exclude images that have both eagles and sparrows) so that the blindspots we induce are non-overlapping and identifiable.

In total, we generated 49 total ECs: 9 ECs had a single blindspot, 20 ECs had 2 blindspots, and 20 ECs had 3 blindspots (selected uniformly at random from the set of all possible pairs and triples).

### H.3 Model Training

Like before, we induce true blindspots by training with the wrong labels for images belonging to blindspots. For each EC, we train a pretrained (on ImageNet) Resnet-18 and perform model selection using the validation set loss. We follow the same process as the COCO experiments (Section G.3) and validate that each model learns the intended blindspots by calculating its recall gap inside vs. outside the blindspot. From the initial 56 ECs that we generated, we successfully induced all of the true blindspots for 48 of them (98%). We retained 9 out of 9 ECs with a single blindspot, 19 out of 20 ECs with two blindspots, and 20 out of 20 ECs with three blindspots.

Finally, we run each BDM using the same hyper-parameters as those detailed in Section 5.

### H.4 Results

**Overall Performance**. We report the average DR across different BDMs with precision and recall thresholds $\lambda_p = 0.5$ and $\lambda_r = 0.3$ in Table 15. We observe the same relative performance ranking as the MS-COCO experiments: `PlaneSpot` achieves performance competitive with past BDMs. All three other BDMs significantly outperform Barlow.

| Method | DR |
|---|---|
| Barlow | 0.39 (0.05) |
| Spotlight | 0.56 (0.04) |
| Domino | 0.57 (0.05) |
| **PlaneSpot** | **0.61** (0.05) |

Table 15: (*OpenImages*) The average DR and its standard error for the pool containing all 49 OpenImages ECs. Each blindspot is defined as a different type of bird.

# I   Image Representation Ablation

In our experiments, we observe that Domino has a higher DR than Spotlight on COCO data (Table 4), which differs from the result observed in the synthetic data experiments (Table 2) where Spotlight had a higher DR than Domino. To better understand this discrepancy, we began by comparing the design choices made by Domino and Spotlight (Table 1). Domino and Spotlight use different image representations and clustering algorithms.

We decided to run a simple ablation experiment where we replace the *image representation* used by Domino with the image representation that is used by Spotlight. We denote this modified version of Domino as `Domino-Model`. Instead of using CLIP, `Domino-Model` runs Domino's core clustering methodology on the model's penultimate layer activations. We chose to ablate the image representation because we hypothesize that CLIP embeddings are relatively better at capturing semantic similarity on real photographic images compared to synthetic images. One plausible explanation for this difference is that photographs from COCO may be more similar to CLIP's training dataset, which consists of text-image pairs scraped from the web (Radford et al., 2021).

| Method | DR | FDR |
|---|---|---|
| Spotlight (Unmodified) | 0.79 (0.03) | 0.09 (0.01) |
| Domino (Unmodified) | 0.64 (0.04) | 0.07 (0.01) |
| `Domino-Model` | 0.80 (0.03) | 0.02 (0.01) |

Table 16: Average BDM DR and FDR with their standard errors across 100 `SpotCheck` ECs.

**Results.** `Domino-Model` achieves a DR competitive with Spotlight (DRs 0.80 vs. 0.79 respectively) on `SpotCheck`. This finding supports our hypothesis that Domino's relatively low performance on `SpotCheck` (compared to its relatively high performance on COCO) is partially attributable to its image representation. More broadly, our ablation illustrates how some BDM design choices may be more or less appropriate for different types of data. In practice, certain image representations may be more or less appropriate for different application domains.

## J    Observations from a Qualitative Inspection of Real Data

In Appendix F, we used the 2D scvis embedding of the model's learned representation as an analysis tool to determine that these BDMs' tendency to merge true blindspots is a failing of the methods themselves. In this section, we do something similar and qualitatively inspect that 2D embedding for models trained on real image datasets in order make additional observations that may be relevant to the design of future BDMs.

**A low-dimensional embedding can work for real data.** In the next subsections, we demonstrate that this 2D embedding contains the information required to identify a wide range of blindspots from prior work and to identify novel blindspots. As a result, it seems that dimensionality-reduction is a potentially important aspect of BDM design or that an interactive approach based on this embedding could be effective.

**Getting a maximally general definition of a specific blindspot often requires merging multiple groups of images.** For example, reaching the conclusions presented in Figures 16, 23, 34, and 31 all required merging several distinct groups of images. As a result, it seems that being able to suggest candidate merges is a potentially important goal for BDM design.

**A small blindspot may appear as a a small cluster (Figure 22) or as a set of outliers (Figure 40).** The former is likely to cause problems for BDMs that use more standard supervised learning techniques (*e.g.,* those that use Linear Models or Decision Trees for their hypothesis class). The later is likely to cause problems for BDMs that model the data distribution (*e.g.,* those that use Gaussian Kernels for their hypothesis class). As a result, it seems that being able to identify blindspots of varying sizes and densities is a potentially important and challenging goal for BDM design.

**The boundaries of a blindspot can be "fuzzy" in the sense that there may be images that do not belong to the blindspot that are both close, in the image representation space, to images in the blindspot and that have higher error.** For example, looking at Figure 20, we can see that the model's confidence decreases relatively smoothly as we approach the region that reflects "baseball gloves with people, but not in a baseball field." However, if we look at the region around this region (Figure 21), we see that these images are almost all in baseball fields. As a result, it seems that being able to handle this fuzziness is a potentially important and challenging goal for BDM design.

### J.1 Preliminaries

Before presenting the details of our results, there are two questions to address.

**What blindspots do we study?** We consider blindspots discovered by prior works (Hohman et al., 2019; Sohoni et al., 2020; Singh et al., 2020; Plumb et al., 2022; Singla et al., 2021) as well as novel blindspots (Adebayo et al., 2022). Note that these prior works:

- Are not all explicitly about blindspots (*e.g.,* some consider "spurious patterns" which imply the existence of blindspots).
- All consider a model that was trained with standard procedures (*e.g.,* no adversarial training) on a real dataset (*e.g.,* no semi-synthetic images or sub-sampling the dataset).
- All consider blindspots that pertain to a single class.

**How are our visualizations produced?** To start, we use scvis (Ding et al., 2018) to learn a 2D embedding of the model's representation of all of the training images of the class of interest. Because scvis combines the objective functions of tSNE and an AutoEncoder, we expect similar images to be near each other and dissimilar images to be far from each other in this 2D embedding. Then, we visualize that embedding and color code the points by the model's confidence in its predictions for the class of interest. As a result, blindspots should appear as groups of lighter colored points.

With this visualization, we start by first exploring what types of images were in different parts of the embedding. Then, given a specific type of image, we go back and highlight the part of the embedding that best represents that type of image. As a result, many of the figures we show are more like "summaries that support our conclusions" than "exhaustive descriptions of all the types of images we identified."

### J.2 Summit

In Figure 1 of their paper, Hohman et al. (2019) show that their method reveals that the model is relying on "hands holding fish" to detect "tench."

- Dataset: ImageNet
- Class: tench
- Blindspots: tench without people holding them

Results of inspecting the 2D embedding:

- The embedding also reveals this blindspot. However, our analysis suggests that this blindspot is too broadly defined because "people holding tench" (Figure 11) has similar performance to "tench in a net" (Figure 12) and that the performance drop comes from "tench underwater" (Figure 13). We confirmed hypothesis by finding images on Google Images that match these descriptions and measuring the model's accuracy on them.
- Additionally, the embedding reveals mislabeled images (Figure 14).

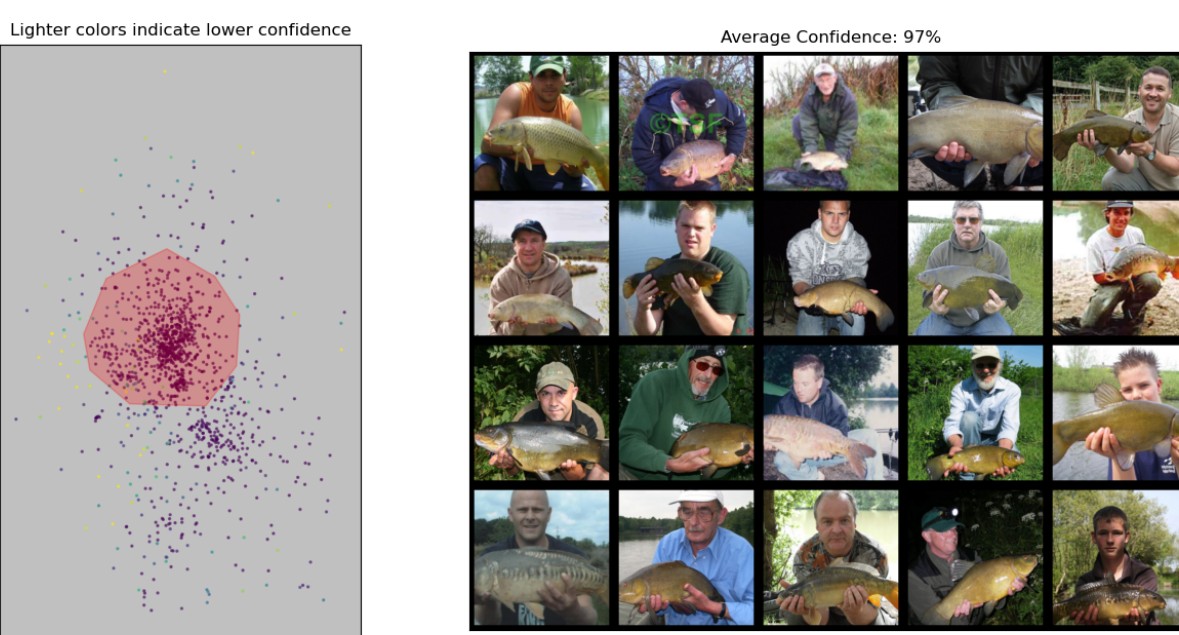

Figure 11: "tench with people holding them." Model accuracy on images from Google Images: 100% (40/40).

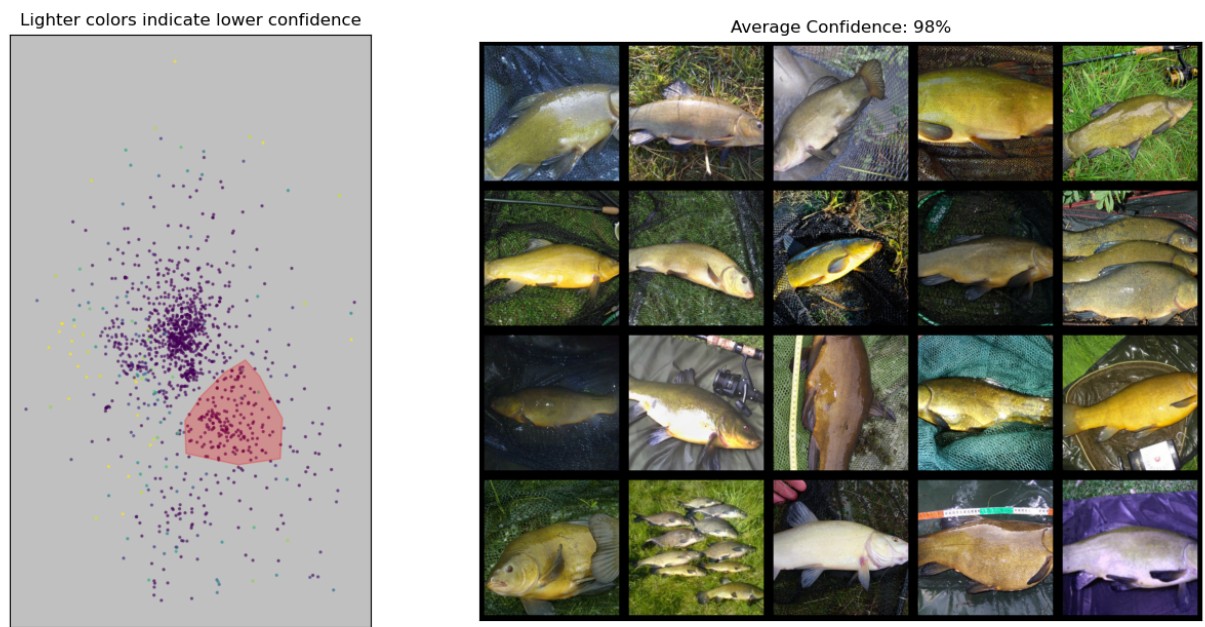

Figure 12: "tench in a net." Model accuracy on images from Google Images: 98% (39/40).

Class: tench

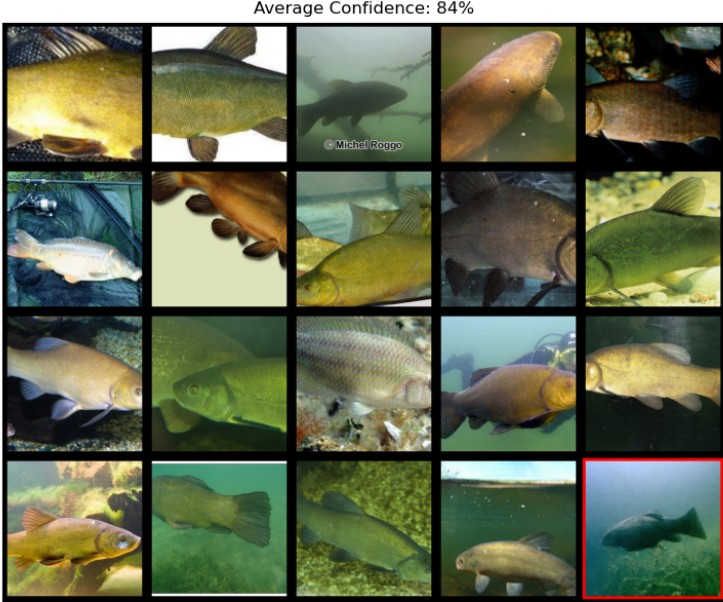

Figure 13: "tench underwater." Model accuracy on images from Google Images: 88% (35/40).

Class: tench

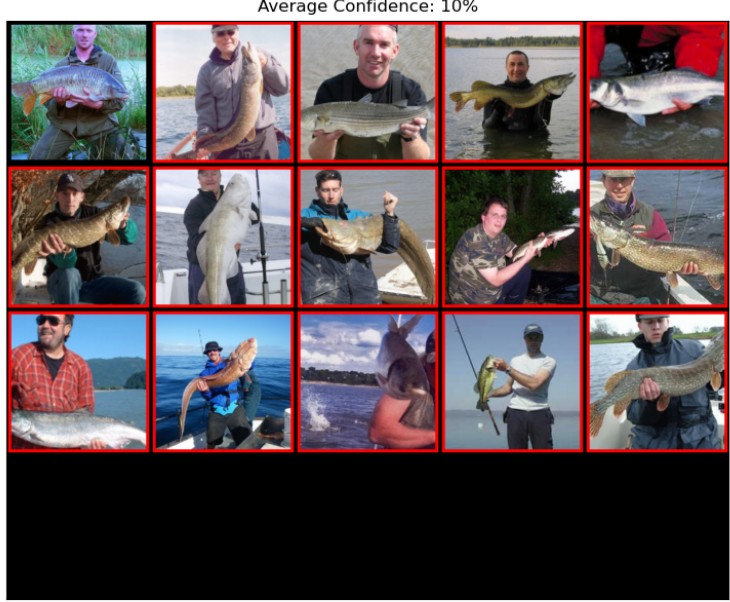

Figure 14: Mislabeled images

### J.3 George

In Appendix C.1.4 of their paper, Sohoni et al. (2020) discuss how their method is not effective at identifying one of the blindspots commonly studied in Group Distributionally Robust Optimization without using an external model's representation for the image representation.

- Dataset: CelebA
- Class: blond hair
- Blindspot: men with blond hair

Results of inspecting the 2D embedding:

- By comparing Figure 15 to Figure 16, we can see that the embedding clearly reveals this blindspot.
- Additionally, Figure 17 shows that the embedding reveals an additional "blond women playing sports" blindspot.

Class: Blond+Hair

Lighter colors indicate lower confidence

Average Confidence: 38%

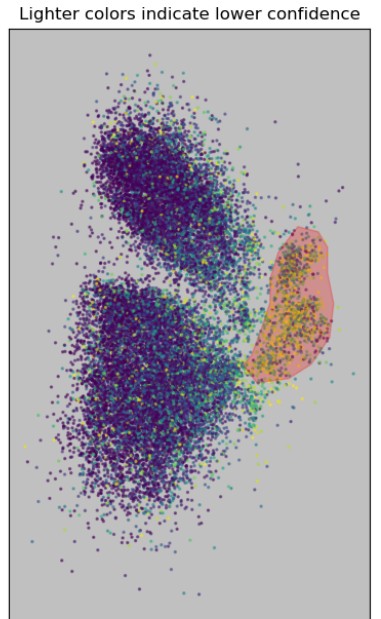

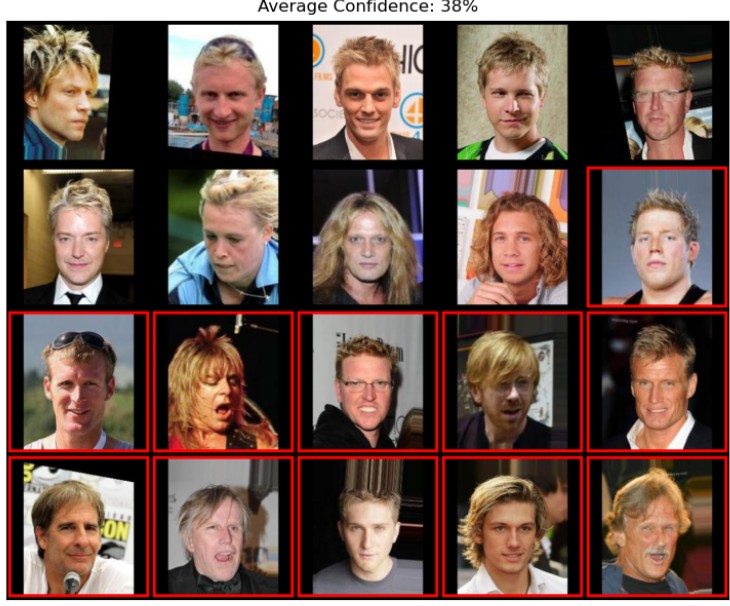

Figure 15: "blond men"

Class: Blond+Hair

Lighter colors indicate lower confidence

Average Confidence: 82%

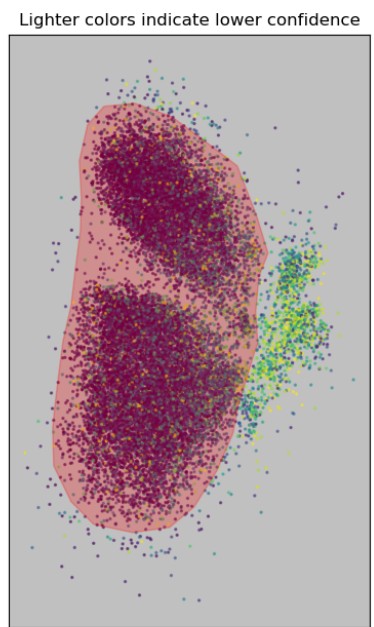

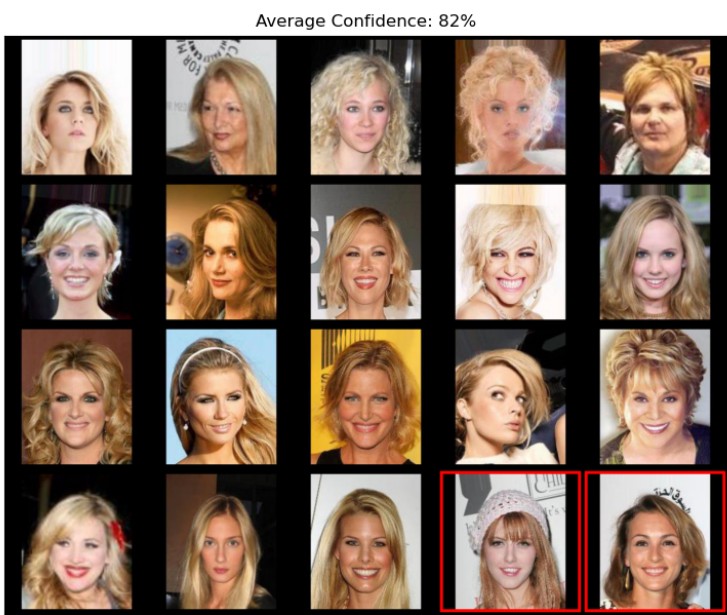

Figure 16: "blond women"

Figure 17: "blond women playing sports"

### J.4 Feature Splitting

In Table 6 of their paper, Singh et al. (2020) organize the spurious patterns that they identify by their own measure of bias. We consider both the most and least biased spurious patterns from that table.

**Most biased.**

- Dataset: COCO
- Class: baseball glove
- Blindspot: baseball gloves without people

Results of inspecting the 2D embedding:

- Comparing Figure 18 to Figure 19, we see that the embedding also reveals this blindspot.
- However, we can also see that the model's performance is not uniform across different types of images of "baseball gloves with people." In particular, the embedding reveals two other blindspots (Figures 20 and 22)

**Least biased.**

- Dataset: COCO
- Class: cup
- Blindspot: cups without dining tables

Results of inspecting the 2D embedding:

- Comparing Figure 23 to Figure 24, we see that the embedding also reveals this blindspot.
- However, we can also see that the model's performance is not uniform across different types of "cups without dining tables." Figures 25, 26, and 27 show a few examples of the additional blindspots the embedding reveals.
- These results highlight an interesting subtlety. The cups in images with dining tables take up a greater portion of the image than the cups in images of sports stadiums, which raises the question: "Does this blindspot exist solely because the model is relying on 'context' that it shouldn't be? Or is the standard image pre-processing making the cups too small (in terms of their size in pixels) to detect?"

Class: baseball+glove

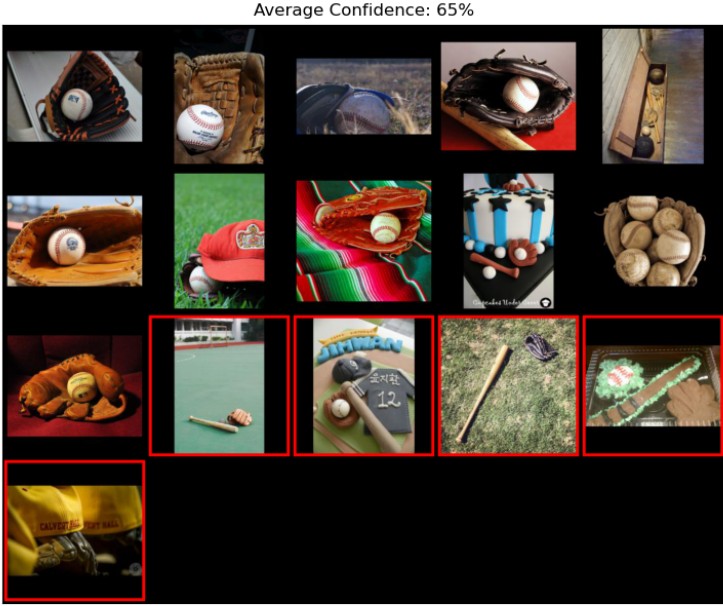

Figure 18: "baseball gloves without people"

Class: baseball+glove

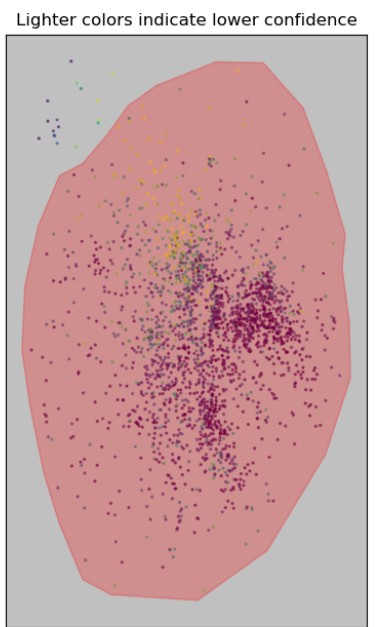 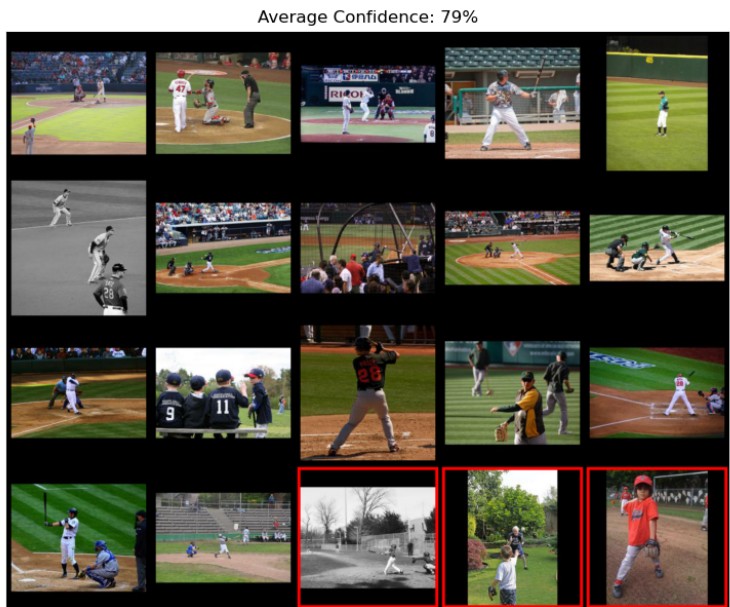

Figure 19: "baseball gloves with people"

Class: baseball+glove

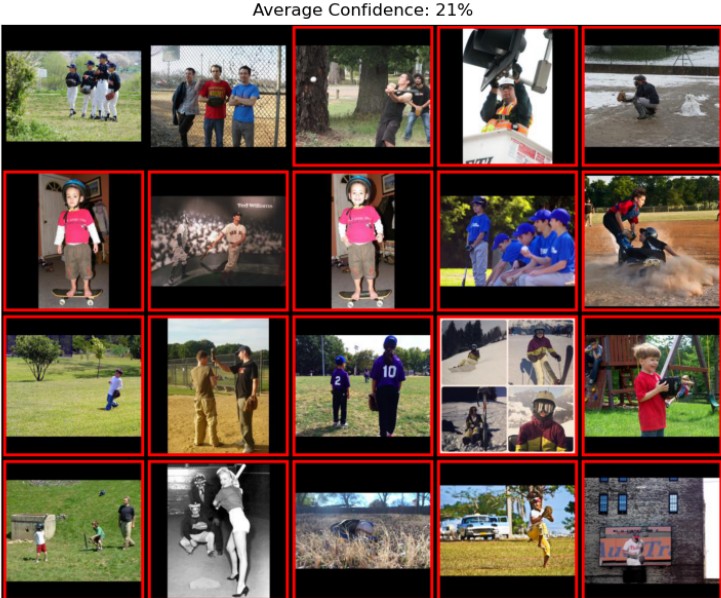

Figure 20: "baseball gloves with people, but not in a baseball field"

Class: baseball+glove

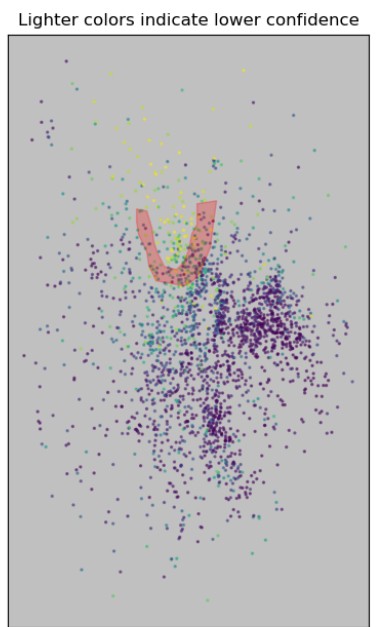
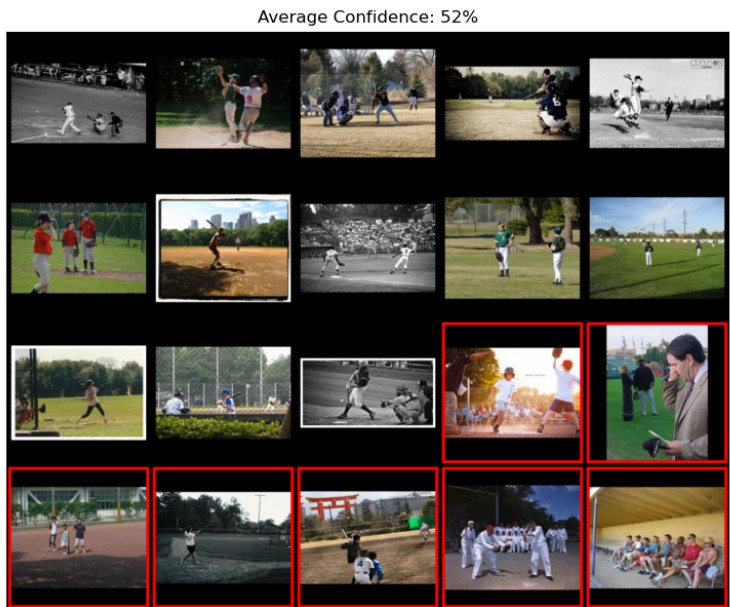

Figure 21: If we look at the region around the "baseball gloves with people, but not in a baseball field" region (Figure 20), we see that the model's confidence is still lower despite the fact that these images still are in baseball fields.

Class: baseball+glove

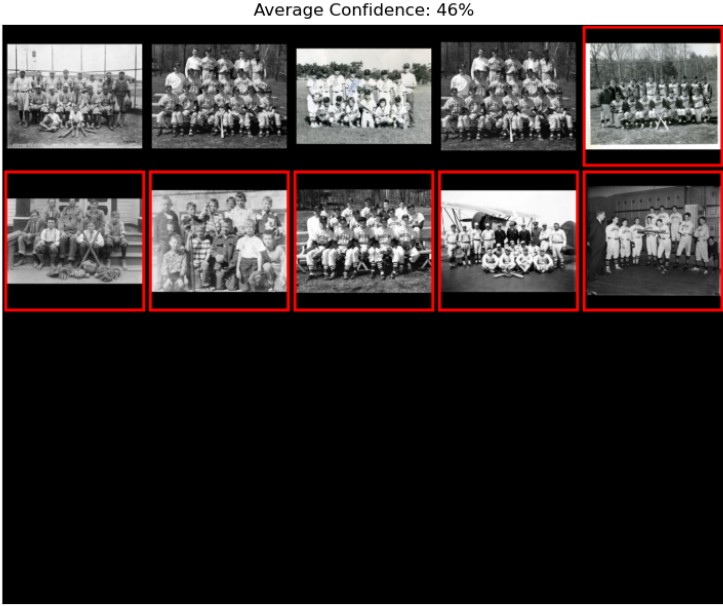

Figure 22: "black and white images of baseball teams"

Class: cup

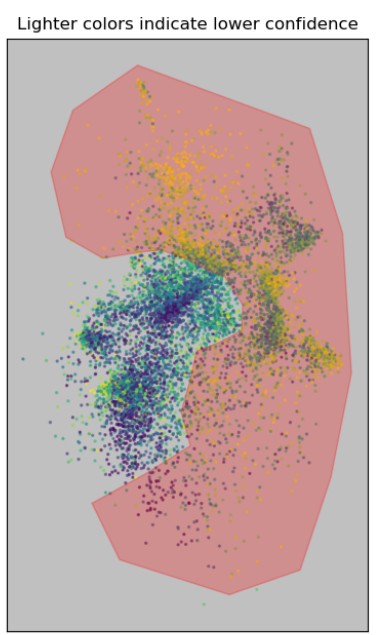

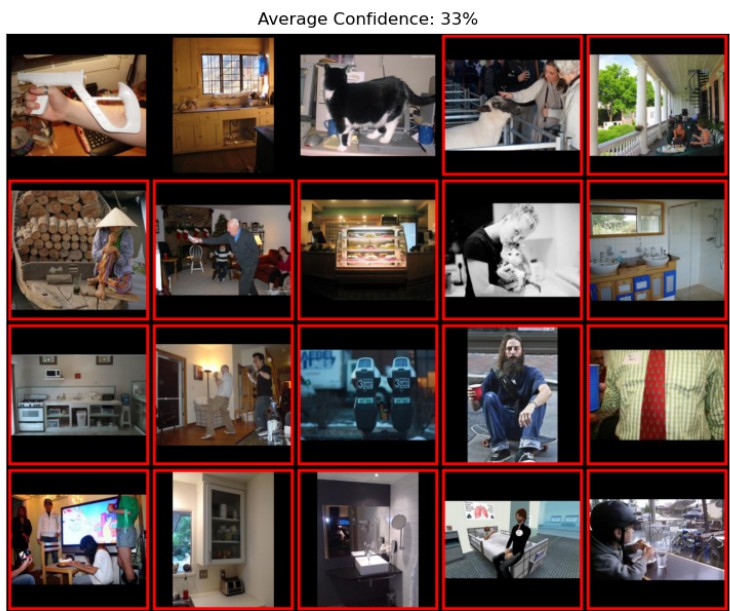

Figure 23: "Cups without dining tables"

Class: cup

Lighter colors indicate lower confidence

Average Confidence: 63%

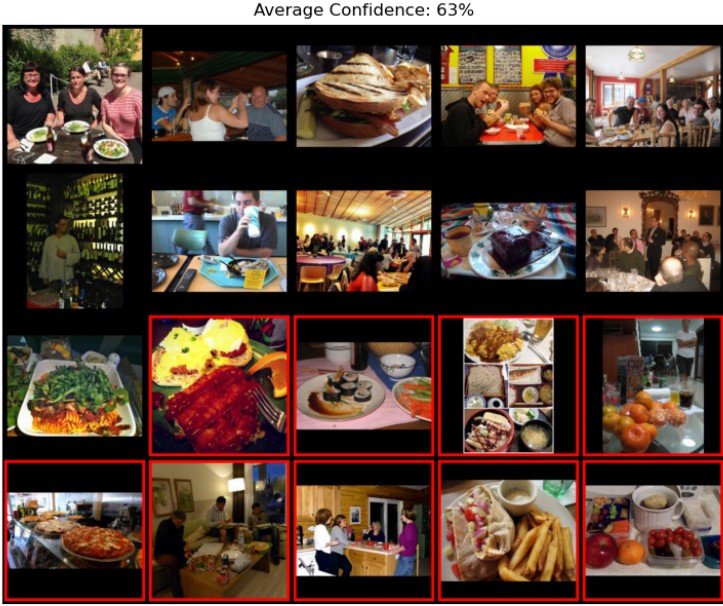

Figure 24: "Cups with dining tables"

Class: cup

Lighter colors indicate lower confidence

Average Confidence: 26%

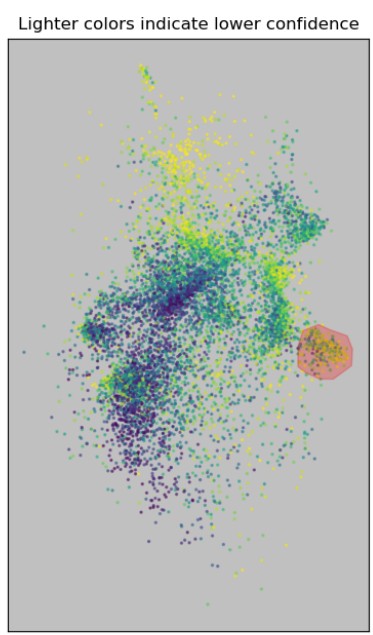

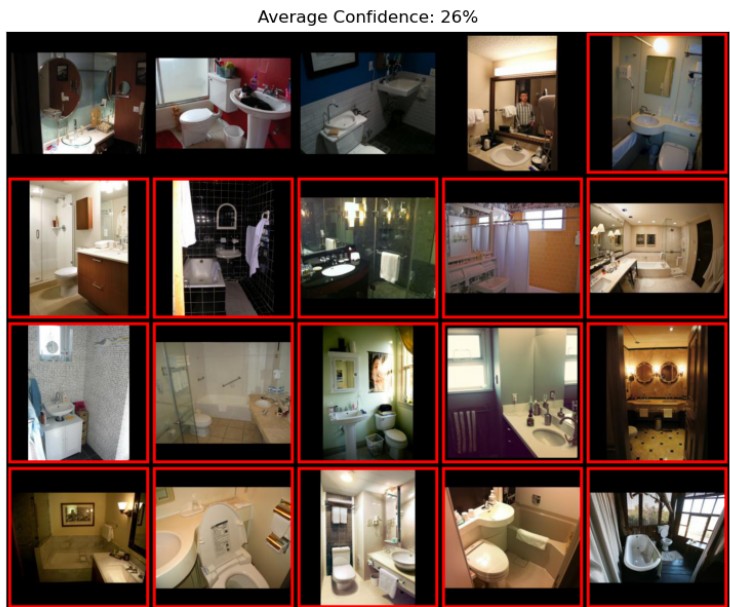

Figure 25: "Cups in bathrooms"

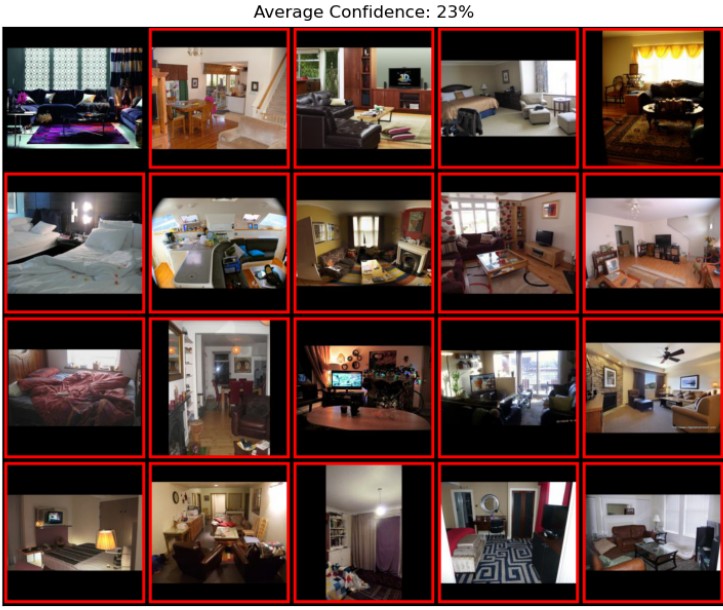

Figure 26: "Cups in living rooms or bedrooms"

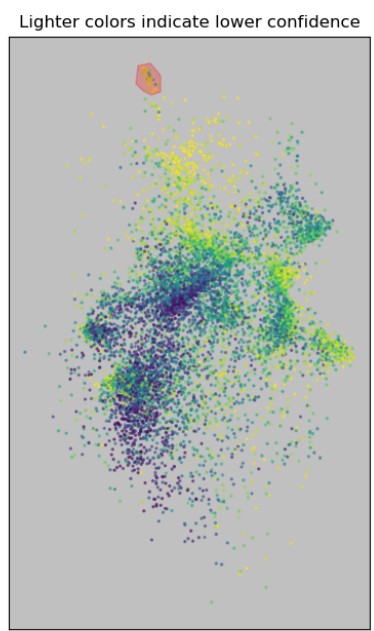
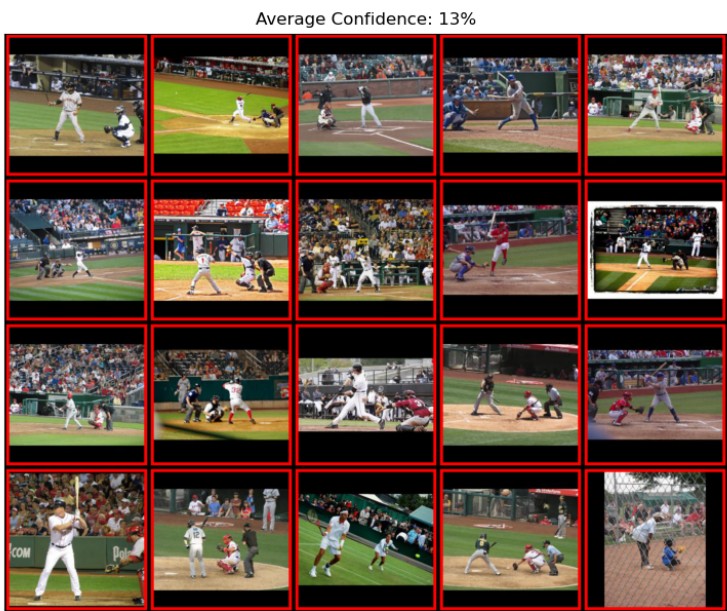

Figure 27: "Cups in sports stadiums". Is the model using 'context' that it shouldn't be? Or is the image pre-processing making detecting these cups very hard by compressing them to only a few pixels? Or is this a limitation of the model architecture?

### J.5 Barlow

In Table 2 of their paper, Singla et al. (2021) summarize some of the blindspots that they identify. We consider two of them. The first is "hog", which is the class with the largest ALER-BER score (which they use as a measurement of how important a blindspot is). The second is "tiger cat", which is the class with the largest increase in error rate for images in the blindspot.

**Largest ALER-BER Score.**

- Dataset: ImageNet
- Class: hog
- Blindspot: not "pinkish animals"

Results of inspecting the 2D embedding:

- By comparing Figure 28 to Figure 29, we can see that the embedding reveals this blindspot.

**Largest increase in error rate.**

- Dataset: ImageNet
- Class: tiger cat
- Blindspot: not "face close up"

Results of inspecting the 2D embedding:

- It does not appear that the embedding reveals this blindspot.
- By comparing Figure 30 to Figures 31 and 32, we can see that the embedding reveals a "grey tiger cats inside" blindspot.
- Looking at Figure 33, we can see that the embedding reveals mislabeled images.

Class: hog

Lighter colors indicate lower confidence

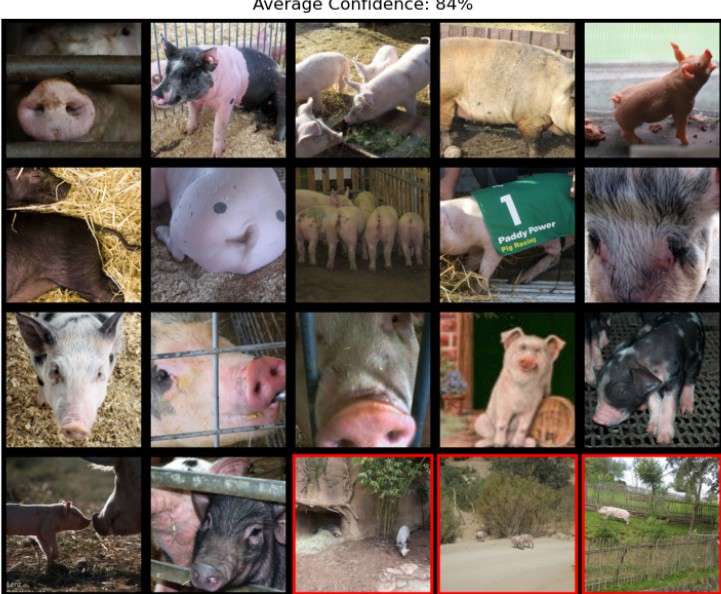

Figure 28: "pink hogs"

Class: hog

Lighter colors indicate lower confidence

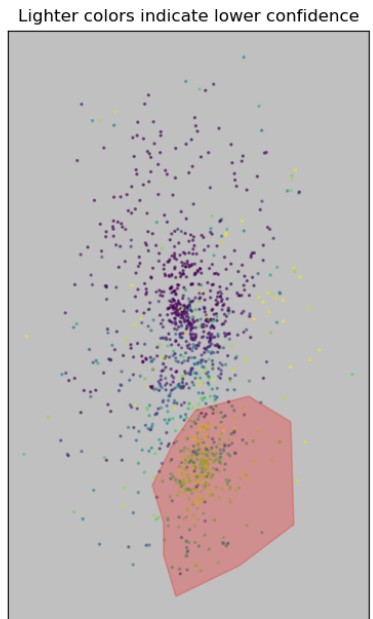
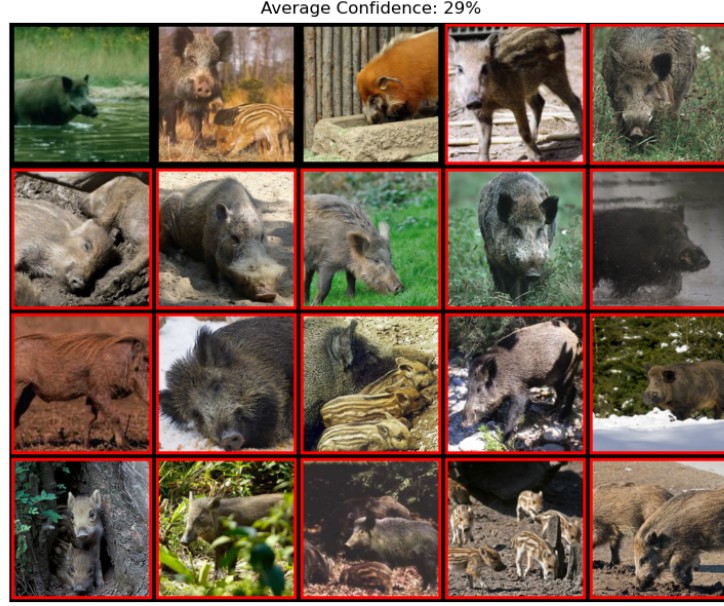

Figure 29: "brown hogs"

Class: tiger_cat

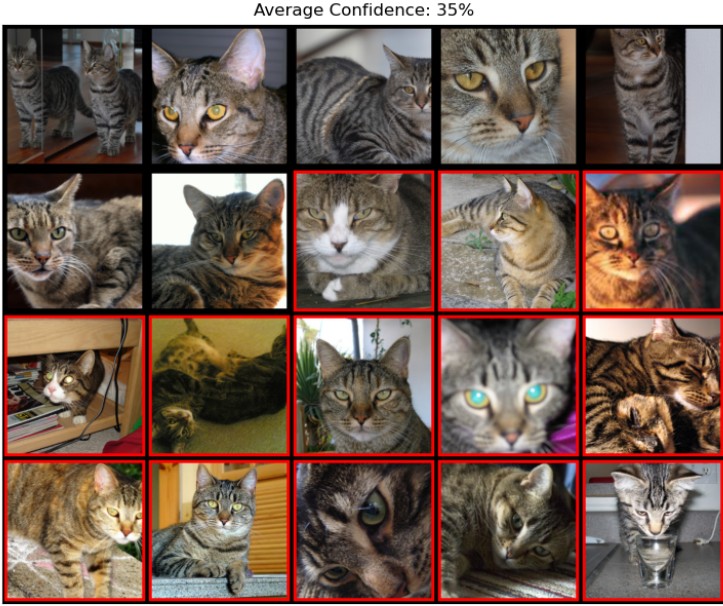

Figure 30: "grey tiger cats inside"

Class: tiger_cat

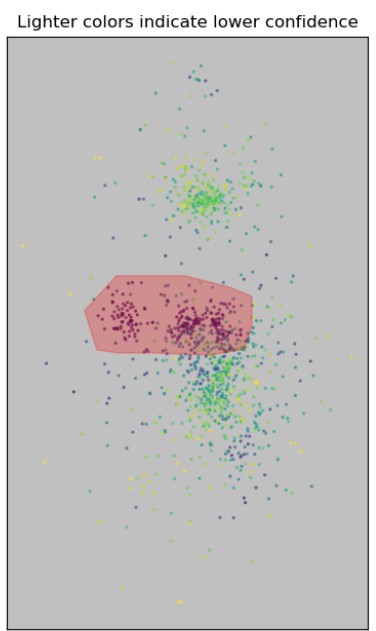
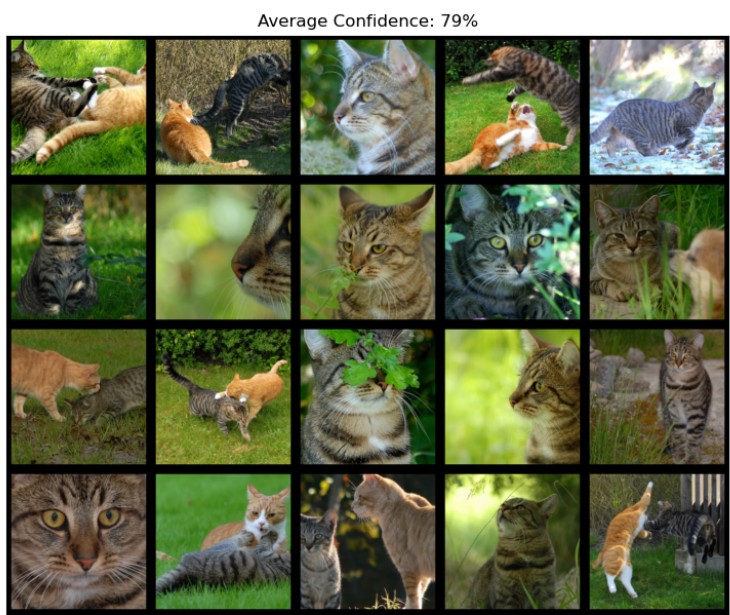

Figure 31: "grey tiger cats outside"

Class: tiger_cat

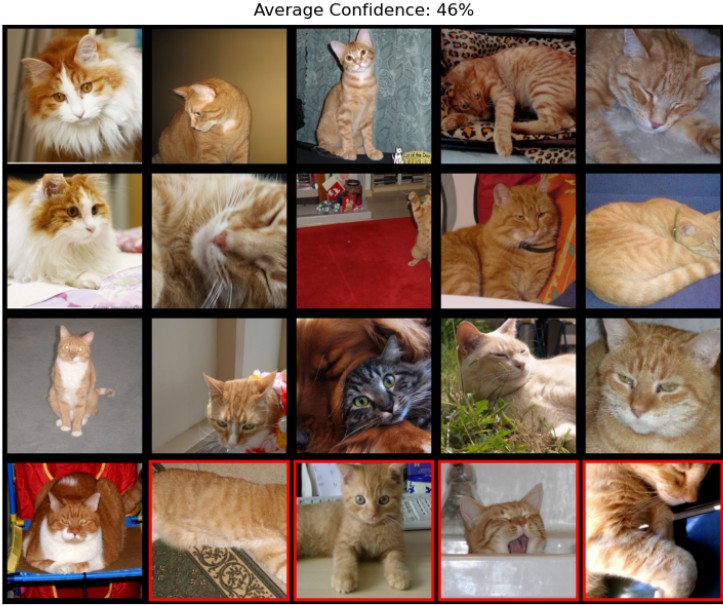

Figure 32: "orange tiger cats"

Class: tiger_cat

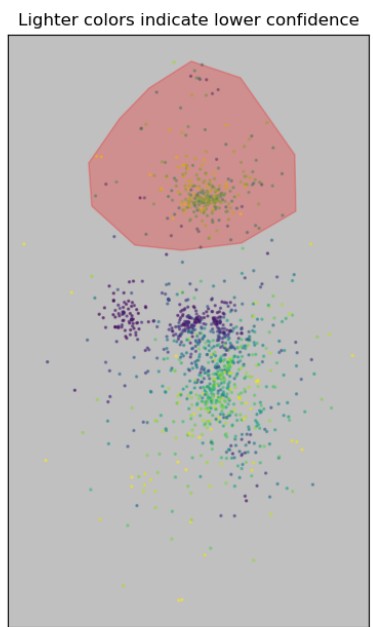
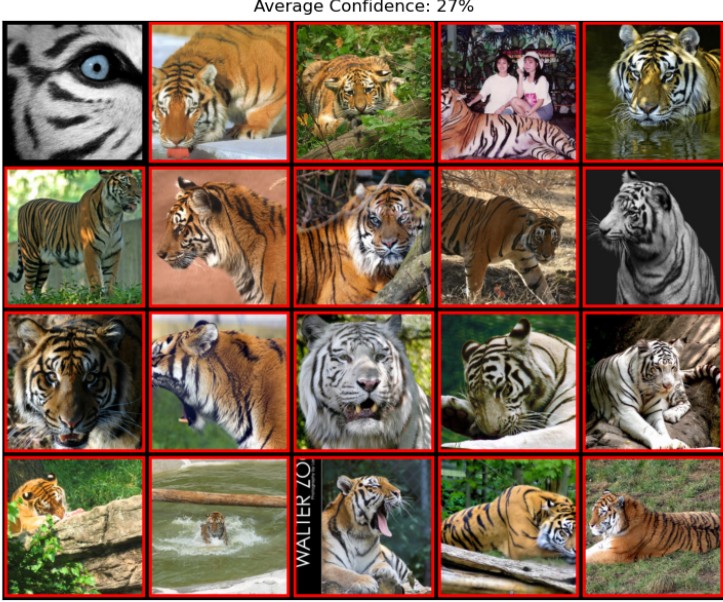

Figure 33: Mislabeled Images

### J.6 Spire

Plumb et al. (2022) claim that, compared to past works on identifying spurious patterns, that theirs is the first to identify negative spurious patterns. So we explore the example of a negative spurious pattern that they give. Additionally, they note that this class is part of two spurious patterns.

- Dataset: COCO
- Class: tie
- Blindspot: ties without people
- Blindspot: ties with cats

Results of inspecting the 2D embedding:

- Comparing Figure 34 to Figure 35, we see that the embedding reveals the "ties without people" blindspot.
- Comparing Figure 36 to Figure 37, we see that the embedding reveals the "ties with cats" blindspot. Note that this is a subset of the "ties without people" blindspot.
- Figures 38 and 39 show a few examples of the additional blindspots the embedding reveals.

Class: tie

Lighter colors indicate lower confidence

Average Confidence: 43%

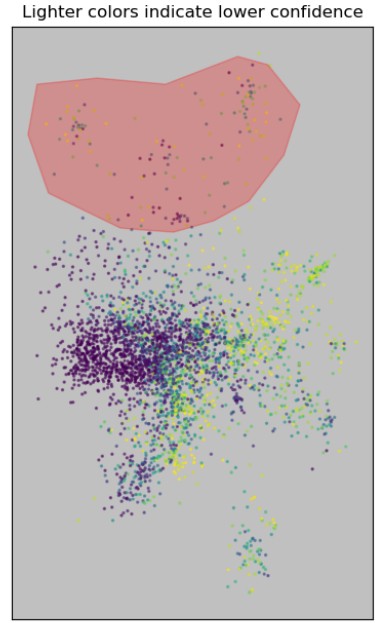

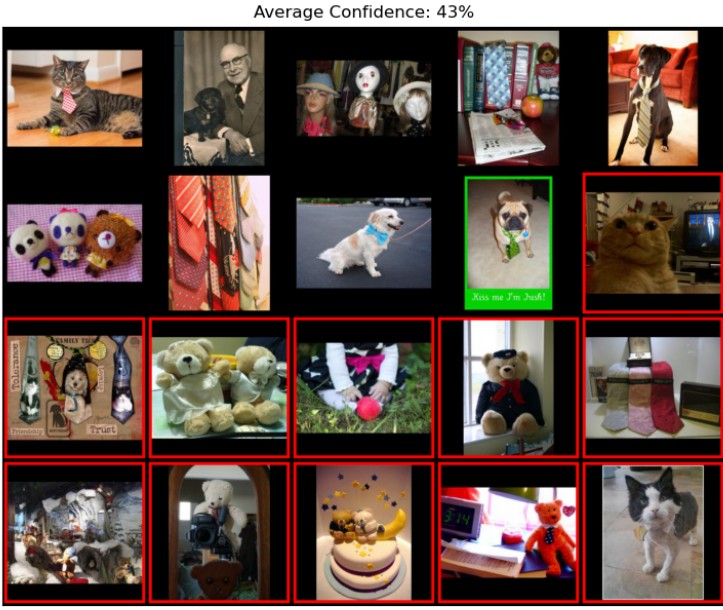

Figure 34: "ties without people"

Class: tie

Lighter colors indicate lower confidence

Average Confidence: 65%

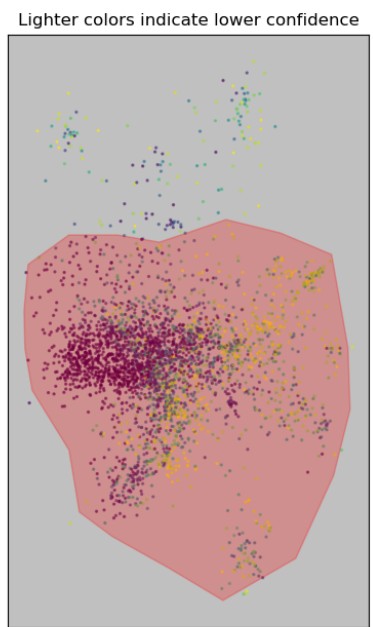

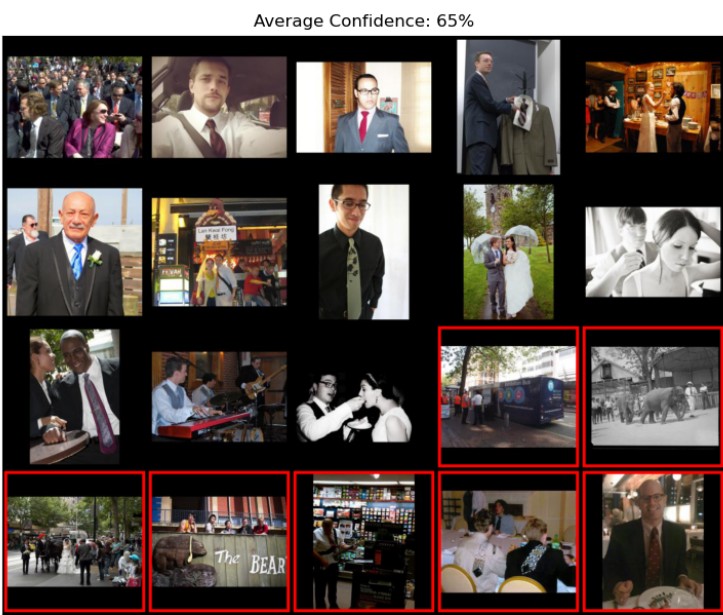

Figure 35: "ties with people"

Class: tie

Lighter colors indicate lower confidence

Average Confidence: 34%

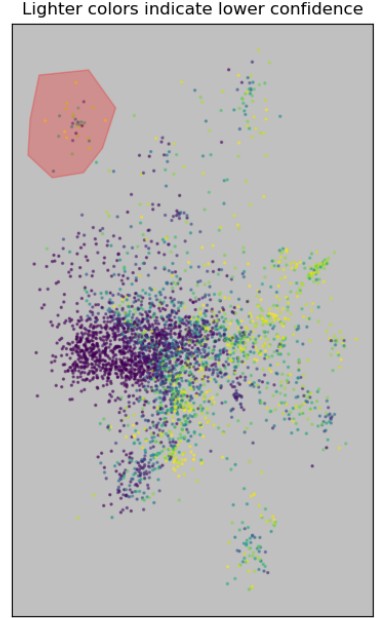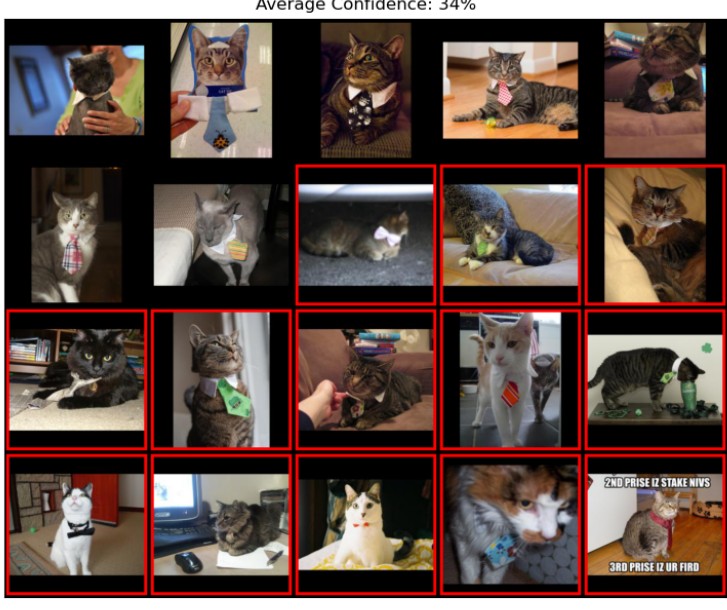

Figure 36: "ties with cats"

Class: tie

Lighter colors indicate lower confidence

Average Confidence: 65%

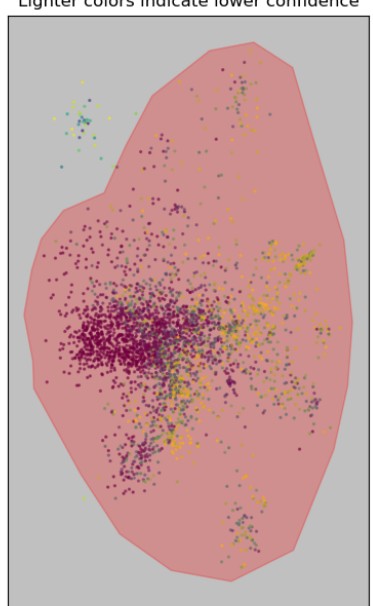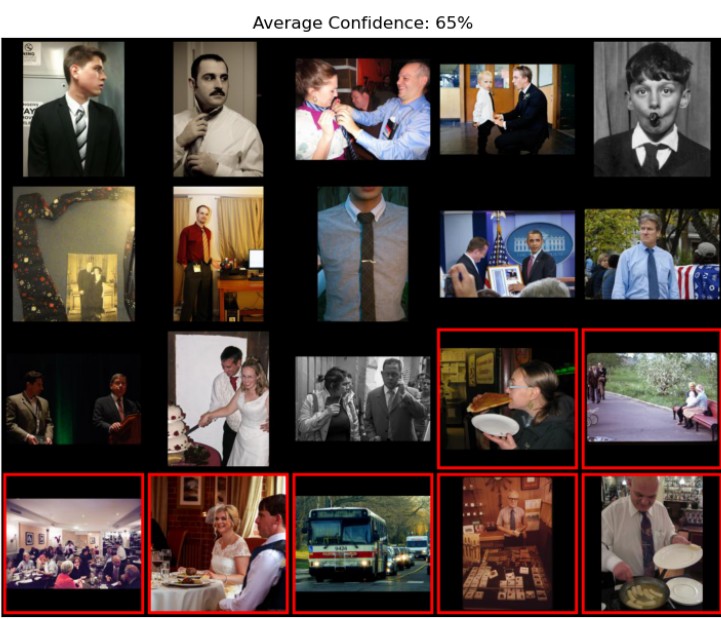

Figure 37: "ties without cats"

Class: tie

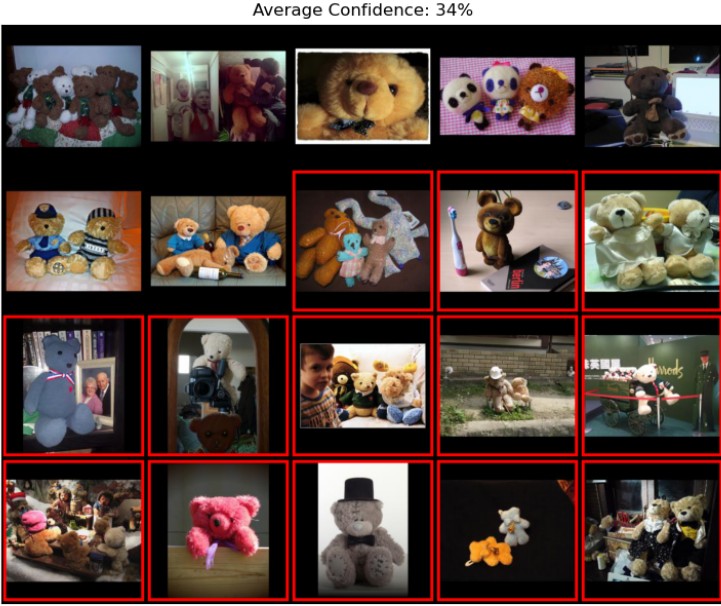

Figure 38: "ties with teddybears"

Class: tie

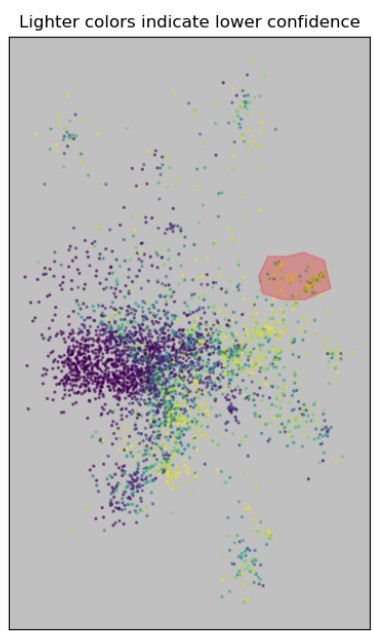
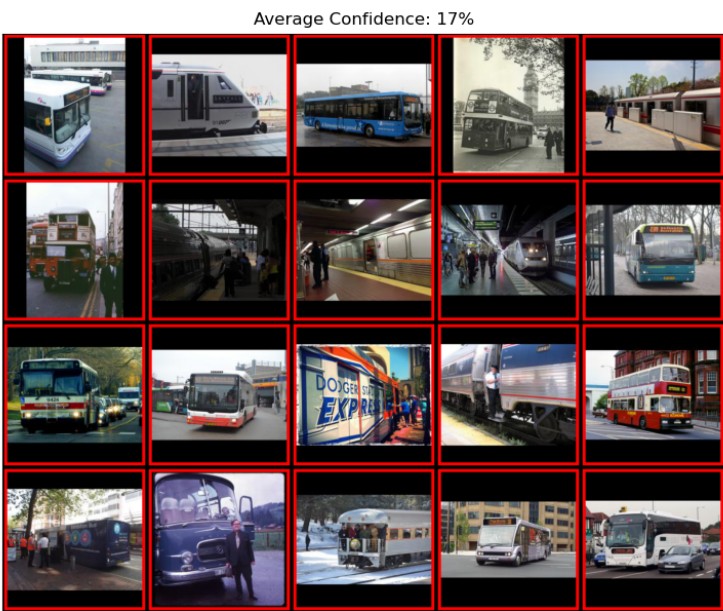

Figure 39: "ties with buses" Is the model using 'context' that it shouldn't be or is the image pre-processing making detecting these ties very hard (the bus driver is frequently wearing a tie, but that tie is very small)?

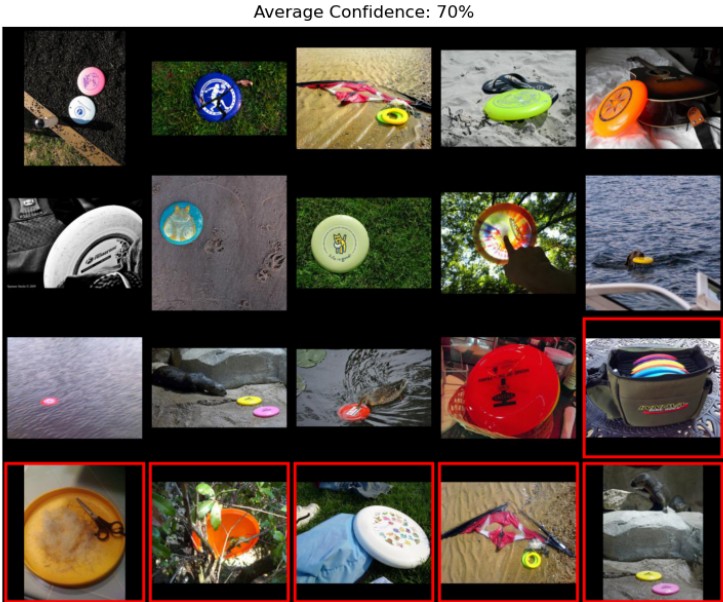

Figure 40: Additionally, when we see that the embedding also reveals the "frisbees without dogs or people" blindspot from (Plumb et al., 2022). However, these images look more like outliers than a proper cluster.

### J.7 New Blindspots

Because it has been observed that it is easier to find "bugs" in models when we know what to look for (Adebayo et al., 2022), we also explore classes that have not been discussed in prior work. We asked a colleague to pick a class from the list of ImageNet classes and they chose "library" because they thought that it seemed related to or correlated with other ImageNet classes.

- Dataset: ImageNet
- Class: library
- Blindspot: images of the outside of libraries
- Blindspot: images of a single bookshelf

Results of inspecting the 2D embedding:

- Figure 41 shows that the model is less able to label an image as a library when the image is of the outside of a library. Interestingly, this is even true when the building is labeled as a library.
- By comparing Figure 42 to Figures 43 and 44, we see that, without the context clues of "people" or "multiple bookshelves" the model is less able to label an image as a library when the image contains only a single bookshelf. This may be a data labeling problem because "bookcase" is another class in ImageNet.

Class: library

Lighter colors indicate lower confidence

Average Confidence: 18%

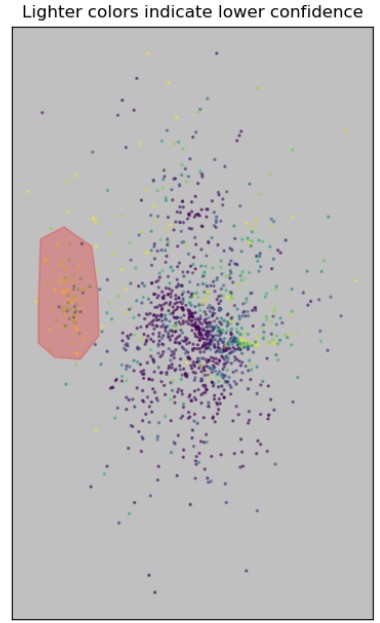

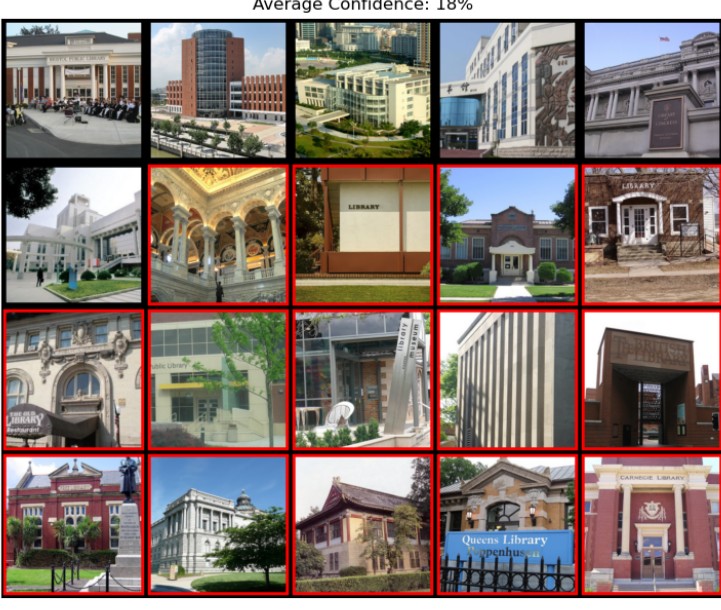

Figure 41: "outside of libraries"

Class: library

Lighter colors indicate lower confidence

Average Confidence: 45%

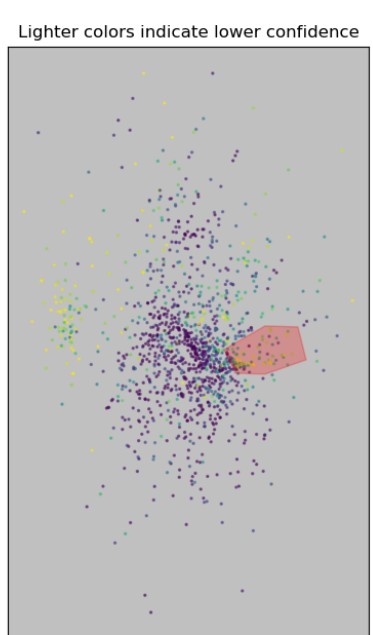

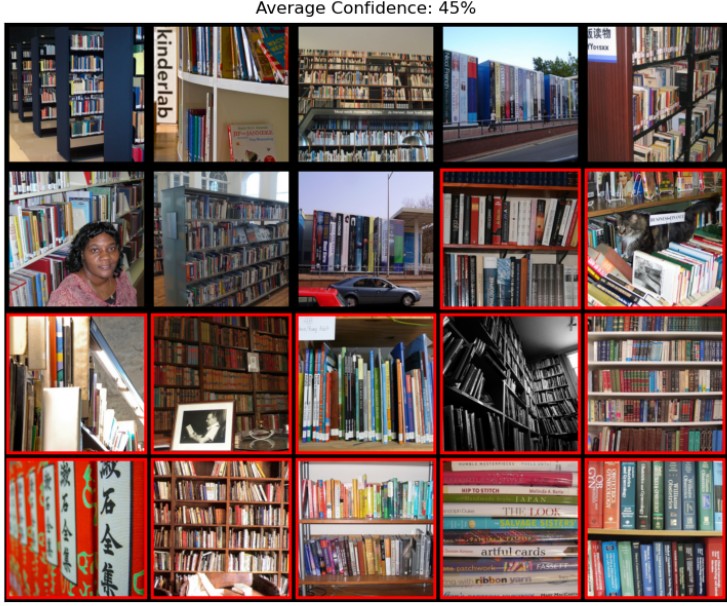

Figure 42: "single bookshelf"

Class: library

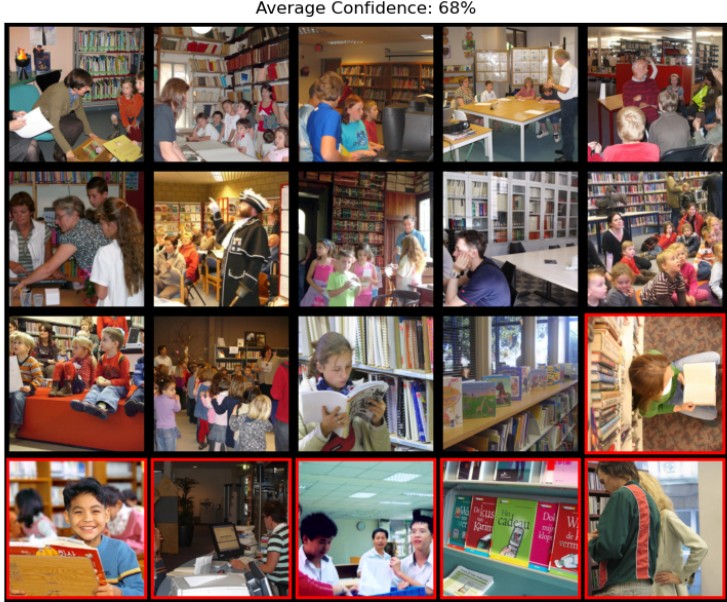

Figure 43: "people near bookshelves'

Class: library

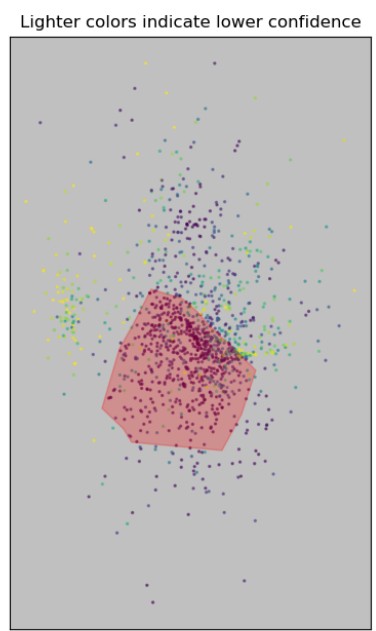
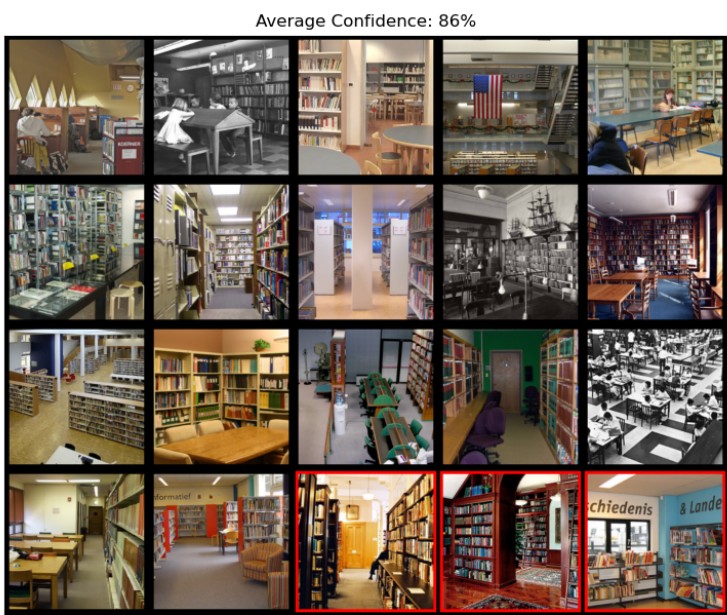

Figure 44: "multiple bookshelves without people"

# K   Evaluating BDMs without Knowledge of the Model's True Blindspots

In this work, we evaluate BDMs by directly comparing the hypothesized blindspots that they return to the model's true blindspots. However, this does not help us evaluate BDMs in real settings, where we do not know the model's true blindspots. In this section, we address this limitation by describing a simple (if labor intensive) process for evaluating BDMs without such knowledge.

Recall from Section 2, that there are two problems with existing approaches for evaluating BDMs without knowledge of the model's true blindspots. Specifically, they do not consider whether or not:

1. A hypothesized blindspot is coherent (*e.g.,* a random sample of misclassified images has high error but may not match a single semantically meaningful description).
2. The model's performance on a hypothesized blindspot is representative of the model's performance on other similar images (*e.g.,* suppose that $f$ has a 90% accuracy on images of "squares and blue circles"; then, by returning the 10% of such images that are misclassified, a BDM could mislead us into believing that $f$ has a 0% accuracy on this type of image).

Additionally, there is a third, more general, problem:

3. These evaluations only work for BDMs and cannot be used to compare BDMs to other methods (*e.g.,* methods from explainable machine learning). Largely, this is because they are designed around the fact that BDMs return hypothesized blindspots as sets of images.

To address these problems, we propose a two-step user-based evaluation process:

1. Using the output of the method, the user writes a text description that describes the images that belong to their hypothesized blindspot.
   - This address Problem 1 because, if the hypothesized blindspots returned by a BDM are not coherent, the user will not be able to describe them.
   - This addresses Problem 3 because the user is converting the output of the method into a standardized format.
   - In Appendix J, we essentially essentially did this step ourselves for the figure captions.
2. We gather new images that match that text description and use those images to measure the model's performance on the user's hypothesized blindspot.
   - This addresses Problem 2 because these newly gathered images were chosen without knowledge of the model's performance on them and are chosen only based on the user's text description.
   - Note that these images could come from an external source or they could be taken from the same set of images that the method was given.
   - In Appendix J.2, we did this step ourselves with images from Google Images.

