# OpenReview forum: "Towards a More Rigorous Science of Blindspot Discovery in Image Classification Models"
_TMLR — Accepted by TMLR_

### Review · Reviewer_uR5h · 2023-04-28

**Summary Of Contributions:**

The authors propose a blindspot discovery framework to evaluate models that aim to find systematic errors in an image classification model. The framework is designed such that blindspots can be identified exhaustively and uniquely to allow rigorous analysis of different methods on toy datasets. The authors then apply their work on real data though they note that this is challenging and some issues may be confounded. The authors compare various methods (Barlow, Spotlight, Domino) with their approach (planespot) and find that their method is both more robust with more blindspots and more robust when the blindspot is defined by multiple features.

**Audience:**

Yes

**Claims And Evidence:**

Yes

**Requested Changes:**

Please see the weakneses above. Explaining / clarifying points 0, 1 in the paper are most important. I also think that the analysis of 4 would be interesting and 2/3 would add to the narrative of the paper.

**Strengths And Weaknesses:**

Strengths:
1. I appreciate trying to formalise and simplify the problem to allow for concrete evaluation metrics and comparisons between multiple methods. The authors construct a toy setting which allows them to evaluate both coverage of blind spots and the ability of a model to identify the 'complete' blind spot as opposed to a subset.
2. Their approach seems to perform better than similar methods on their toy dataset.
3. The authors work on generalising to a larger dataset while also mentioning the limitations (e.g. that there are many confounding factors as you try such an approach on a larger dataset which makes taking concrete conclusions challenging). However the results seem to generalise to this setting and also presents a way to evaluate such methods on a real dataset reasonably robustly.

Weaknesses:
0. I'm confused a bit on how the blind spots are computed. Is the analysis run on the full training set of images?
1. My understanding is the authors finetune a model with 0-label for iamges that are 'in the blind spot'. Does this make sense? Shouldn't the model just not be fine-tuned on them (this is more representative of what actually happens in the real world). Or is it that these images are held out? The text seems to imply the latter but figures the former (e.g. Fig 1). If the former, I wonder if any of the conclusions would hold for a standard classifier as maybe it wouldn't be distributed the same so the methods would perform differently.
2. Generalizability to other tasks. While interesting to look at classification, do these techniques generalise to other settings (e.g. generation, regression tasks, etc.). As more and more of ML looks at larger models trained on vast amounts of data, how does this analysis hold up and can it still be useful. I think at least a discussion around this would be useful.
3. Missing related work. There are a few works looking at finding missing regions (e.g. blind spots, e.g. failures) in classification models using generative models or datasets that are not referenced by the authors. I think these would add to the paper:
- Adaptive Testing of Computer Vision Models. Gao et al
- Discovering Bugs in Vision Models using Off-the-shelf Image Generation and Captioning. Wiles et al.
4. If my interpretation of 0 is correct, an interesting analysis would be, how many examples do you need of blind spots in the train set in order to find them. Presumably with 1 you wouldn't but with more (maybe 10?, 100?) you would?

---

> ### Author Response · Authors · 2023-05-25
> **Response to Reviewer uR5h, Part 1**
>
> Thank you for your review and suggestions to improve the work!  We respond to your listed weaknesses below.
>
> ----
>
> **Point #0**: "*I'm confused a bit on how the blind spots are computed. Is the analysis run on the full training set of images?*"
>
> **Response**:
> * In our experiments, we run each BDM to discover blindspots on each **test set**.
> * We chose to run the BDMs on a hold-out test set (instead of the train set) because the model's performance on the test set is more likely to be representative of model performance during deployment, especially given that over-parameterized deep learning models often achieve near-perfect train accuracy [1].
> * We updated the manuscript to make our motivation for this choice more clear in Section 5.1.
>
> ----
>
> **Point #1**: "*My understanding is the authors finetune a model with 0-label for images that are 'in the blind spot' … Shouldn't the model just not be fine-tuned on them (this is more representative of what actually happens in the real world)."... "If the former, I wonder if any of the conclusions would hold for a standard classifier as maybe it wouldn't be distributed the same so the methods would perform differently.*"
>
> **Response**:
> * Yes, as stated in Section 3.1, we induce the true blindspots by fine-tuning the model using a train dataset where the blindspots are mislabeled.  Please see **Point #1** in our General Response, where we further explain why we chose this method and discuss additional experiments where we tried alternative methods (including the reviewer's suggestion of holding out images) to induce the true blindspots.
> * We agree with the reviewer's point that a "standard classifier" is trained on an unmodified dataset, which differs from the datasets in our experiments because 100% label noise on each blindspot is unlikely to occur in practice.  We dedicate a paragraph to discussing this limitation in the manuscript (Section 6.2).
> * Overall, we agree with the reviewer that further exploration of *alternative methods to induce blindspots* is worthwhile.  However, we believe that understanding when (and why) a model will learn an intended blindspot is beyond the scope of this work (because it is closely related to the problem of model generalization), and could be aided by our framework.
>
> ----
>
> **Point #2**:  "*Generalizability to other tasks: While interesting to look at classification, do these techniques generalise to other settings (e.g. generation, regression tasks, etc.)*"
>
> **Response**:
> * We agree with the reviewer that discovering systematic errors in settings beyond classification is an important research direction.  In our work, we state explicitly that we study the problem of blindspot discovery for *image classification* (Section 2), an assumption shared by the BDMs that we evaluate.
> * Existing BDMs for image classification optimize for finding coherent, *low-accuracy* slices of data.  To transfer these approaches from a classification to a regression or generation setting, we first need to specify some measure of "accuracy".  For generation settings, defining such a measure is not straightforward:  how can we measure the quality of a generated image?  Therefore, we believe that transferring existing or developing novel BDMs for such settings is an important direction for future work that is out-of-scope for our paper.  We hope that the tools we've developed to evaluate BDMs for classification may be a good starting point to evaluate tools proposed for other settings.
> * We updated the manuscript's title to "Towards a More Rigorous Science of Blindspot Discovery in Image ___Classification___ Models" to make our work's scope more clear.
>
> ----
>
> **Point #3**: "*Missing related work: There are a few works looking at finding missing regions (e.g. blind spots, e.g. failures) in classification models using generative models or datasets that are not referenced by the authors.*"
>
> **Response**:  Thank you for the references!
>
> * Our manuscript already cites AdaVision (Gao et al.) in the list of "human-in-the-loop" methods in Section 2.  While related, AdaVision differs from the BDMs we study because it requires a human-in-the-loop (to manually label the expected model behavior on the retrieved examples).
> * We updated the manuscript to add a reference to Wiles et al. in Section 2.  While related, their method differs from the BDMs we study because they assume access to a model that can generate realistic images from text prompts (e.g. DALL-E).  This assumption may be limiting in practice, as realistic image generation is computationally expensive and may not be possible in specialized domains (see AdaVision's discussion of Wiles et al. in their "Related Work" section).

---

> ### Author Response · Authors · 2023-05-25
> **Response to Reviewer uR5h, Part 2**
>
> (continued)
>
> **Point #4**: "*... an interesting analysis would be, how many examples do you need of blind spots in the train set in order to find them.*"
>
> **Response**: Thanks for the suggestion!
>
> * Studying how the number of examples in each blindspot affects BDM performance is part of the motivation for our "specificity" analyses (Figure 3).
> * In our synthetic data experiments, we vary the likelihood $p(x)$ of sampling an image belonging to each blindspot by varying "the number of semantic features that define the blindspot", where each additional semantic feature makes the blindspot 50% less likely.  On average, the blindspots defined with 5, 6, and 7 semantic features had 230, 94, and 56 test set images respectively.
> * We found that more specific blindspots (with fewer examples in the test set input to BDMs) are on average harder for BDMs to find.
>
> Please let us know if we misinterpreted your suggestion.
>
> ----
>
> References:
>
> [1] V Feldman, C Zhang.  "What Neural Networks Memorize and Why: Discovering the Long Tail via Influence Estimation".  Proceedings of the 24th Conference on Neural Information Processing Systems (NeurIPS), 2020.  Accessed via https://arxiv.org/abs/2008.03703
>
> [2] Sabri Eyuboglu, Maya Varma, Khaled Kamal Saab, Jean-Benoit Delbrouck, Christopher Lee-Messer, Jared Dunnmon, James Zou, and Christopher Re. "Domino: Discovering systematic errors with cross-modal embeddings". In International Conference on Learning Representations, 2022. URL https://openreview.net/forum?id=FPCMqjI0jXN.
>
> [3] Joon Sik Kim, Gregory Plumb, and Ameet Talwalkar. "Sanity simulations for saliency methods". In Proceedings of the 39th International Conference on Machine Learning, 2022.

---

### Review · Reviewer_NDYv · 2023-05-09

**Summary Of Contributions:**

This paper proposes a systematic evaluation of blindspot discovery methods. BDMs find and group semantically meaningful subsets of the data on which a model performs worse, to allow for inspection and eventual updating. To this end, this work introduces 1) 2 new benchmark datasets for BDMs, 2) corresponding evaluation metrics (DR and FDR across different blindspot specificities), and 3) a new simple baseline for BDMs that projects features onto 2 dimensions.

**Audience:**

Yes

**Claims And Evidence:**

No

**Requested Changes:**

Have 2 requested changes along the lines of the weaknesses pointed above:

1. (Manuscript-level change). Include more motivation and real-world use cases for why this specific formulation of BDMs makes sense and/or is useful. Currently this is quite undermotivated.

2. (Additional experiments). The main utility of such a benchmark is to test BDMs on real-world data. In this work, results on only 1 dataset, COCO, is presented. To make the claims and results analysis more generalizable, it would be best for this work to have at least 2 datasets to support their claims. OpenImages is one such dataset that contains enough metadata to make this feasible.

#2 is critical to securing my recommendation for acceptance.

**Strengths And Weaknesses:**

Strengths:
- The synthetic data controlled setup is done fairly well, as it allows programmatic control and ability to induce very specific blindspots.
- The relevant baselines are compared against (Table 1).
- Analysis of similarities and differences in the results compared with the COCO dataset is quite nice (eg Domino performing much better as it uses pretrained CLIP encoder)
- New PlaneSpot baseline is very simple and seems to be fairly effective on both synthetic and semi-synthetic dataset.

Weaknesses:
- The motivation for why this specific instantiation of BDMs is useful is lacking. For example, it's not clear what exactly a human would do with a group of unordered images - versus perhaps a text description that more succinctly summarizes the blindspot.
- Limited evaluation suite. The results would be more compelling with more real-world datasets - for example, OpenImages also have a lot of metadata/annotations that would allow building a similar blindspot benchmark as to COCO. Are the results there consistent as well?
- For real-world data, PlaneSpot does not seem to offer much benefit over Domino.

---

> ### Author Response · Authors · 2023-05-25
> **Response to Reviewer NDYv**
>
> Thank you for the review, and for acknowledging strengths of the work!  We took your suggestion and designed an additional experiment using data from OpenImages.  This experiment is currently in-progress, and we hope to share results before the end of the discussion period.
>
> We respond to each of your listed weaknesses & requested changes below.
>
> ----
>
> **Point #1**: "*The motivation for why this specific instantiation of BDMs is useful is lacking … it's not clear what exactly a human would do with a group of unordered images - versus perhaps a text description that more succinctly summarizes the blindspot*".
>
> **Response**:  Past works have proposed two ways that the output groups $\hat{\Psi}$ can be used:
>
> 1. A human stakeholder (such as a model developer or auditor) can look at each group of images to form behavioral hypotheses of the subgroups where the model fails (e.g. from the Domino [blog post](http://ai.stanford.edu/blog/domino/), "the model fails to detect cars in photos of cars taken from the inside") [1, 2, 3].
>     * In Barlow's user study (Section 9), practitioners used the groups of images output by Barlow to imagine possible actions they could take to fix the model, such as targeted data collection [3].
>     * We agree with the reviewer's point that "what exactly a human would do" to make sense of the unordered groups of images is a relatively understudied and important direction for follow-up work.
> 2. The group of points output by the BDM can be given as input to algorithms that aim to "fix" the blindspot by learning an updated model, such as GDRO [4] or other post-processing algorithms [5, 6].
>
> We updated Section 2 to include more motivation for our formulation (Requested Change #1).
>
> ----
>
> **Point #2**: "*The results would be more compelling with more real-world datasets … To make the claims and results analysis more generalizable, it would be best for this work to have at least 2 datasets to support their claims.*"
>
> **Response**:  We've taken your suggestion and are running an additional experiment where we aim to replicate the result from Table 3 using a subset of data from OpenImages.  We hope to share this result before the end of the discussion period.  Thank you for your patience (as we train dozens of ResNets)!
>
> ----
>
> **Point #3**: "*For real-world data, PlaneSpot does not seem to offer much benefit over Domino.*"
>
> **Response**:  We agree with the reviewer that overall, PlaneSpot has a similar average DR to Domino in our COCO experiments (Table 4) – thus, we conclude that PlaneSpot is "competitive with" Domino.
>
> We observed in Figure 6 that PlaneSpot's DR is more robust than Domino's as the number of true blindspots increases.  This result suggests that PlaneSpot may have some relative benefit over Domino in settings where a model has multiple blindspots – a setting that many recent works speculate is actually quite common in practice [7].
>
> ----
> References:
>
> [1]  Sabri Eyuboglu, Maya Varma, Khaled Kamal Saab, Jean-Benoit Delbrouck, Christopher Lee-Messer, Jared Dunnmon, James Zou, and Christopher Re. Domino: Discovering systematic errors with cross-modal embeddings. In International Conference on Learning Representations, 2022. URL https://openreview.net/forum?id=FPCMqjI0jXN.
>
> [2] Greg d’Eon, Jason d’Eon, James R Wright, and Kevin Leyton-Brown. The spotlight: A general method for discovering systematic errors in deep learning models. arXiv preprint arXiv:2107.00758, 2021.
>
> [3] Sahil Singla, Besmira Nushi, Shital Shah, Ece Kamar, and Eric Horvitz. Understanding failures of deep networks via robust feature extraction. In Proceedings of the IEEE/CVF Conference on Computer Vision and Pattern Recognition, pp. 12853–12862, 2021.
>
> [4]  Nimit Sohoni, Jared Dunnmon, Geoffrey Angus, Albert Gu, and Christopher Ré. No subclass left behind: Fine- grained robustness in coarse-grained classification problems. Advances in Neural Information Processing Systems, 33:19339–19352, 2020.
>
>
> [5] Michael P Kim, Amirata Ghorbani, and James Zou. Multiaccuracy: Black-box post-processing for fairness in classification. In Proceedings of the 2019 AAAI/ACM Conference on AI, Ethics, and Society, pp. 247–254, 2019.
>
> [6] Zhang, M., Sohoni, N.S., Zhang, H.R., Finn, C. &amp; Re, C.. (2022). Correct-N-Contrast: a Contrastive Approach for Improving Robustness to Spurious Correlations. Proceedings of the 39th International Conference on Machine Learning.
>
> [7]  Li, Zhiheng & Evtimov, Ivan & Gordo, Albert & Hazırbaş, Caner & Hassner, Tal & Ferrer, Cristian & Xu, Chenliang & Ibrahim, Mark. (2022). A Whac-A-Mole Dilemma: Shortcuts Come in Multiples Where Mitigating One Amplifies Others. 10.48550/arXiv.2212.04825.

---

> > ### Author Response · Authors · 2023-05-27
> > **Re: OpenImages Experiments**
> >
> > We updated the draft to include results from additional real data experiments using data from OpenImages as the reviewer suggested in **Appendix H**.  Our additional experiments trained 49 additional models with blindspots defined using OpenImages' class hierarchy (see details in the Appendix).
> >
> > We observed the same relative performance ranking across BDMs as we did in the MS-COCO experiments (Table 15).  Particularly, PlaneSpot achieves a higher average DR than past BDMs (i.e. Domino and Spotlight).

---

> > > ### Comment · Reviewer_NDYv · 2023-06-08
> > > **Thanks for responses**
> > >
> > > I thank the authors for responding to my review and updating the manuscript. Weakness 1 has been adequately resolved in the updated manuscript. In addition, I appreciate the inclusion of the OpenImages experiments, which are consistent with the results from the COCO experiments. Therefore, Points 2 and 3 have been resolved, though I would suggest the authors add the OpenImages results to the main text for the camera ready.

---

### Review · Reviewer_U7Xx · 2023-05-11

**Summary Of Contributions:**

The paper focuses on Blindspot Discovery Methods (BDM), which expose semantically meaningful subsets of data where the image classifier performs worse. When evaluating BDMs, previous works are limited by having incomplete knowledge of the model’s blindspots. Thus, the paper proposes SpotCheck for BDM evaluation based on synthetic datasets. The authors also propose a new BDM, PlaneSpot, based on dimension reduction. The experiments conducted on synthetic SpotCheck and real-world MS-COCO datasets show that PlaneSpot is competitive and can expose factors that affect BDM performances, e.g., the number of blindspots. Finally, the paper discusses the future directions of BDMs.

**Audience:**

Yes

**Broader Impact Concerns:**

From my perspective, the paper does not have ethical concerns or need to add a Broader Impact statement.

**Claims And Evidence:**

Yes

**Requested Changes:**

### Major requests:
1. Clarification on the novelty of the proposed method compared to Domino.
2. Adding ablation studies of the proposed method regarding dimensionality reduction algorithms and the number of dimensions.
3. Formal definitions of blindspot types—rare, correlation, or noisy labels.

### Minor requests that would strengthen the paper:
1. Clarify the restriction of the assumption about counterfactual image generation.
2. Adding discussions to closely related works.

### Minor suggestions:
I suggest the authors add dataset names to the captions of results tables and figures, e.g., Tables 2-4 and Figures 2-7, which can better guide the readers to differentiate between the synthetic and real-world experiments.


**Strengths And Weaknesses:**

## Strengths

1. The paper identifies an important problem that previous BDM evaluations lack complete knowledge of true blindspots. Thus, the paper is well-motivated to propose SpotCheck based on the synthetic dataset, where complete knowledge of ground-truth blindspots can be obtained. The new evaluation metrics (Section 3.2) are carefully designed.
2. The paper makes a good summary of existing BDMs in terms of three aspects of design choices (Table 1).
3. The experiments are extensive, e.g., 100 experiment configurations (ECs) in Section 5.1.
4. The experiments provide insights into factors that influence BDM performance, e.g., number of blindspots, specificity of blindspots, etc., which are beneficial for future works to develop better BDMs.
5. The paper is well-written and easy to follow.

## Weaknesses

### Primary concerns
1. [Concerns on novelty]: The proposed method, PlaneSpot, has limited novelty compared to Domino. Both methods use dimension reduction and Mixture of Gaussians to perform clustering over image representations. Especially the paper claims that dimensionality reduction methods receive less attention. However, based on the summary in Table 1, didn’t both GEORGE and Domino explore using dimensionality reduction for BDMs?
2. [Lack of ablation study of the proposed method]: There is no ablation study on (1) the dimension reduction algorithm (e.g., scvis vs. PCA) and (2) the number of dimensions (d) for dimension reduction, where the paper only directly uses d=2 without empirical justification.
3. [Lack of clarification of blindspot types]: I think the paper lacks a formal definition of blindspot types. The paper defines it as “systemic errors” (2nd paragraph in Section 1). However, according to Eyuboglu et al., 2022, there are three underlying reasons for systematic errors—rare, correlation, and noisy labels (Section 4.1.1 in (Eyuboglu et al., 2022)). I am confused by which type(s) of systematic error are used in this paper. For the synthetic dataset in SpotCheck, I think the paper defines the blindspot as noisy labels (“training images belonging to a blindspot are mislabeled.” in Figure 1). However, in COCO, the paper seems to use both “rare” (“animal prediction task can have the blindspot zebra” in Appendix E.2) and noisy labels (“wrong labels for images belonging to blindspots” in Appendix E.3) to define the blindspots.




### Minor Concerns
1. [Assumption about producing counterfactual images]: I don’t understand why the authors claim the ability to produce counterfactual images “restrict the applicability of their respective methods” (Problem definition in Section 2). In fact, other methods [1,2] can produce counterfactual images without making assumptions about the blindspots, which are not discussed or cited in the paper.
2. [Missing discussion on related works]: Some closely related work should be cited and discussed.
    * In [2], the authors also design large-scale experimental settings to evaluate bias discovery results based on synthetic datasets, which is closely related to SpotCheck evaluation.
    * Other synthetic evaluation [3,4] frameworks were proposed for the closely related topic of biases and shortcuts.
    * Li et al. [5] found the challenges in detecting multiple biases (i.e., blindspots) of LfF [6], which is closely related to the paper’s finding, i.e., BDMs perform worse in settings with multiple blindspots (Section 5.1.1).


### References

[1] Oran Lang, Yossi Gandelsman, Michal Yarom, Yoav Wald, Gal Elidan, Avinatan Hassidim, William T. Freeman, Phillip Isola, et al., “Explaining in Style: Training a GAN to explain a classifier in StyleSpace,” in ICCV, 2021.

[2] Zhiheng Li and Chenliang Xu, “Discover the Unknown Biased Attribute of an Image Classifier,” in ICCV, 2021.

[3] Elias Eulig, Piyapat Saranrittichai, Chaithanya Kumar Mummadi, Kilian Rambach, William Beluch, Xiahan Shi, and Volker Fischer, “DiagViB-6: A Diagnostic Benchmark Suite for Vision Models in the Presence of Shortcut and Generalization Opportunities,” in ICCV, 2021.

[4] Katherine L. Hermann and Andrew K. Lampinen, “What shapes feature representations? Exploring datasets, architectures, and training,” in NeurIPS, 2020.

[5] Zhiheng Li, Anthony Hoogs, and Chenliang Xu, “Discover and Mitigate Unknown Biases with Debiasing Alternate Networks,” in ECCV, 2022.

[6] Junhyun Nam, Hyuntak Cha, Sungsoo Ahn, Jaeho Lee, and Jinwoo Shin, “Learning from Failure: Training Debiased Classiﬁer from Biased Classiﬁer,” in NeurIPS, 2020.

---

> ### Author Response · Authors · 2023-05-25
> **Response to Reviewer U7Xx, Part 1**
>
> Thank you for your review and suggestions to improve the work!  We're glad that the reviewer believes the paper "*identifies an important problem*" and "*is well-motivated to propose SpotCheck*".  We respond to each of your listed concerns and requested changes below.
>
> ----
>
> **Primary Concern #1**: "*Concerns on novelty: The proposed method, PlaneSpot, has limited novelty compared to Domino.  Both methods use dimension reduction and Mixture of Gaussians … didn't both GEORGE and Domino explore using dimensionality reduction for BDMs?*"
>
> **Response**: Please see **Point #2** in the General Response where we clarify our motivation for developing and benchmarking PlaneSpot, and how PlaneSpot differs from prior work.  Additionally, we note that "novelty of the studied method(s)" is **not** part of the TMLR [acceptance criteria](https://jmlr.org/tmlr/acceptance-criteria.html).
>
> We added clarifications to the Introduction and Related Work sections of the manuscript as the reviewer requested (Major Request #1).
>
> ----
>
> **Primary Concern #2**: "*Lack of ablation study of the proposed method: There is no ablation study on (1) the dimension reduction algorithm (e.g., scvis vs. PCA) and (2) the number of dimensions (d) for dimension reduction.*"
>
> **Response**:  We added results from additional experiments in Appendix D where we ablate the dimensionality reduction method and number of dimensions (Major Request #2).  In summary, we found that scvis consistently outperforms PCA, and performs similarly as tSNE.  We also found that BDM performance does not degrade as we decrease the dimension $d$ to 2.
>
> ----
>
> **Primary Concern #3**: "*Lack of clarification of blindspot types: … According to Eyuboglu et al, there are three underlying reasons for systematic errors – rare, correlation, and noisy labels.  I am confused by which type(s) of systematic error are used in this paper.*"
>
> **Response**:  In our work, we define a blindspot as a property of a model.  As stated in Section 2, a blindspot is a coherent set of images $\Psi$ where the model has worse performance on $\Psi$ than $D \setminus \Psi$.  We do not categorize blindspots based on their "cause" like Eyuboglu et al. do.  In our experiments, we induce true blindspots by training with the wrong labels (i.e. are "noisy label slices").  We further justify why we chose this method in **Point #1** of the General Response.
>
> We updated the manuscript as requested by the reviewer to clarify this choice in Section 3.1 (Major Request #3).

---

> ### Author Response · Authors · 2023-05-25
> **Response to Reviewer U7Xx, Part 2**
>
> (continued)
>
> **Minor Concern #1**:  "*Assumption about producing counterfactual images:  I don't understand why the authors claim the ability to produce counterfactual images 'restricts the applicability of [the] respective methods*'".
>
> **Response**:  Thank you for the references!  We added the two papers to our list of related work.
>
> We agree with the reviewer's point that methods that produce counterfactual images (like those cited) may aid blindspot discovery.  However, we believe that the ability to generate realistic counterfactual images is presently hard or infeasible in some settings.  Despite the recent success of several popular commercial generative models (like Stable Diffusion), generating realistic images remains challenging in domain-specific settings with limited data (especially data that is dissimilar to data found on the web), such as medical imaging [1].  Therefore, we focus our study on methods that do *not* rely on generating realistic data, as they can be applied in a wider range of settings and domains.
>
> ----
>
> **Minor Concern #2**:  "*Missing discussion on related work: some closely related work should be cited and discussed.*"
>
> **Response**: Thank you for the references!  We added references [2, 3] to the Related Work section of our manuscript, added a reference to [4] in Section 6.1, and added references [5, 6] to our Appendix A.1.
>
> ----
>
> **Minor Suggestion**:  "*I suggest the authors add dataset names to the captions of results tables and figures, e.g., Tables 2-4 and Figures 2-7.*"
>
> **Response**: Thanks for the suggestion!  We added the dataset name to each table and figure caption.
>
> ----
>
> [1]  DuMont Schütte, A., Hetzel, J., Gatidis, S. et al. Overcoming barriers to data sharing with medical image generation: a comprehensive evaluation. npj Digit. Med. 4, 141 (2021). https://doi.org/10.1038/s41746-021-00507-3
>
> [2] Oran Lang, Yossi Gandelsman, Michal Yarom, Yoav Wald, Gal Elidan, Avinatan Hassidim, William T. Freeman, Phillip Isola, Amir Globerson, Michal Irani, and Inbar Mosseri. Explaining in style: Training a GAN to explain a classifier in stylespace. CoRR, abs/2104.13369, 2021. URL https://arxiv.org/abs/ 2104.13369.
>
> [3]  Z. Li and C. Xu, "Discover the Unknown Biased Attribute of an Image Classifier," in 2021 IEEE/CVF International Conference on Computer Vision (ICCV), Montreal, QC, Canada, 2021 pp. 14950-14959.
>
> [4] Zhiheng Li, Anthony Hoogs, and Chenliang Xu. 2022. Discover and Mitigate Unknown Biases with Debiasing Alternate Networks. In Computer Vision – ECCV 2022: 17th European Conference, Tel Aviv, Israel, October 23–27, 2022, Proceedings, Part XIII. Springer-Verlag, Berlin, Heidelberg, 270–288. https://doi.org/10.1007/978-3-031-19778-9_16
>
> [5] Eulig, Elias et al. “DiagViB-6: A Diagnostic Benchmark Suite for Vision Models in the Presence of Shortcut and Generalization Opportunities.” 2021 IEEE/CVF International Conference on Computer Vision (ICCV) (2021): 10635-10644.
>
> [6] Hermann, Katherine L. and Andrew Kyle Lampinen. “What shapes feature representations? Exploring datasets, architectures, and training.” ArXiv abs/2006.12433 (2020).

---

> > ### Comment · Reviewer_U7Xx · 2023-05-28
> > **Concerns Addressed**
> >
> > I appreciate the authors' response.
> >
> > First, let me apologize for using the word "novelty." My original concern was the similarity between the proposed PlaneSpot and the previous Domino method. Now I think the response and the added ablation study on the dimension reduction algorithm in Appendix D addressed my concern.
> >
> > Other responses well-addressed my remaining concerns, e.g., clarification of the blindspot type, generating counterfactual images, adding related works, etc.
> >
> > Overall, I think this is a good paper on blindspot discovery. I recommend accepting the paper.

---

### Author Response · Authors · 2023-05-25
**General Response**

We would like to thank the reviewers for their thoughtful feedback and suggestions to strengthen our work!  We followed reviewers' recommendations to run several additional experiments that we believe address the reviewers' concerns.  We also updated our manuscript (**see highlighted changes**) in response to reviewers' comments.

We respond to related comments in a general response.

---

**Point #1: Method used to induce the true blindspots (Reviewer U7Xx and uR5h)**

* As stated in Section 3.1, for our experiments we induce true blindspots by fine-tuning a pretrained model using a dataset where *all images that belong to a blindspot are mislabeled*.

* This method (training with mislabeled images to induce true blindspots) is shared by several prior studies, including Domino (the other existing quantitative evaluation framework for BDMs) [1] and Kim et al. [2].

* Reviewer uR5h suggested that we explore **alternative methods to induce true blindspots**, such as "holding out" images belonging to the blindspot during training.

* We decided to **run additional experiments** detailed in **Appendix B** where we varied the method used to induce the true blindspots for 50 SpotCheck ECs.  We tried two additional methods:
    1. Excluding all images that belong to each true blindspot during training (Reviewer uR5h's suggestion)
    2. Varying the label noise rate (we tried noise rates 30% and 50%)
* In summary, we observed that **none of the alternative methods that we tried were as successful at inducing all of the true blindspots**.  In contrast, training with mislabeled images had a 100% success rate at inducing the true blindspots.
* As discussed in Section 5.2, failing to learn all of the intended blindspots can cause undesirable confounding effects.
* In conclusion, we chose to train with mislabeled images as it was the only reliable method we found that consistently induced the desired blindspots.  **We added this clarification to Section 3.1 of the manuscript.**

---

**Point #2: Motivation for PlaneSpot (Reviewer U7Xx and NDYv)**

* Our primary motivation when designing PlaneSpot was to benchmark a simple BDM that clusters on a **2D** image representation.  We were specifically interested in studying a 2D representation for its visualization potential for *interactive, human-in-the-loop* blindspot discovery (see Section 6.3).

* Past works (such as Domino and GEORGE) have explored applying dimensionality reduction to the image representation.  However, their approaches differ significantly from PlaneSpot's:
    * Domino uses a much larger representation (d = 128) that cannot be easily visualized [1].
    * Like our work, the authors of GEORGE propose using a low-dimensional embedding because clustering is generally easier in low dimensions. However, their proposed BDM selects a different number of UMAP components for different datasets (see discussion in their Appendix B.3.4) [3].
    * In contrast, we were specifically interested in studying the effectiveness of clustering from a 2D embedding.
    * **We added a clarifying footnote to Section 2.**

---

References:

[1]  Sabri Eyuboglu, Maya Varma, Khaled Kamal Saab, Jean-Benoit Delbrouck, Christopher Lee-Messer, Jared Dunnmon, James Zou, and Christopher Re. Domino: Discovering systematic errors with cross-modal embeddings. In International Conference on Learning Representations, 2022. URL https://openreview.net/forum?id=FPCMqjI0jXN.

[2] Joon Sik Kim, Gregory Plumb, and Ameet Talwalkar. Sanity simulations for saliency methods. In Proceedings of the 39th International Conference on Machine Learning, 2022.

[3] Nimit Sohoni, Jared Dunnmon, Geoffrey Angus, Albert Gu, and Christopher Ré. No subclass left behind: Fine- grained robustness in coarse-grained classification problems. Advances in Neural Information Processing Systems, 33:19339–19352, 2020.

---

### Author Response · Authors · 2023-07-11
**Camera ready version added**

We've uploaded a final camera-ready version of the paper to OpenReview where we added affiliations and links to code.

Thank you so much all for your time reviewing our work!

---

### Decision · Action_Editors · 2023-06-16

**Recommendation:** Accept as is

**Comment:**

The authors do a thorough job of justifying why their approach, SpotCheck, is a rigorous framework for evaluating blindspot detection methods. Their experiments are insightful and will be valuable knowledge for readers of TMLR interested in blindspot discovery.

The reviewers were in agreement that the paper should be accepted post rebuttal.

**Audience:**

Yes, blindspot discovery has been a major challenge for highly performant deep learning models, that are inaccurate in specific cases. However, although several approaches have been proposed for the same, a systematic evaluation framework has not been developed, and the authors' work addresses this need.

**Claims And Evidence:**

The authors develop a novel framework, SpotCheck, to evaluate blindspot discovery methods that identify semantically meaningful regions where an image classifier is inaccurate. The authors's claims regarding SpotCheck are well supported by strong experiments and questions that the reviewers had were well addressed in the rebuttal.